# Flexible theta sequence compression mediated via phase precessing interneurons

**Angus Chadwick[1,2‡], Mark CW van Rossum[1†], Matthew F Nolan[3*†]**

[1]Institute for Adaptive and Neural Computation, School of Informatics, University of Edinburgh, Scotland, United Kingdom; [2]Neuroinformatics Doctoral Training Centre, School of Informatics, University of Edinburgh, Edinburgh, United Kingdom; [3]Centre for Integrative Physiology, University of Edinburgh, Edinburgh, United Kingdom

**Abstract** Encoding of behavioral episodes as spike sequences during hippocampal theta oscillations provides a neural substrate for computations on events extended across time and space. However, the mechanisms underlying the numerous and diverse experimentally observed properties of theta sequences remain poorly understood. Here we account for theta sequences using a novel model constrained by the septo-hippocampal circuitry. We show that when spontaneously active interneurons integrate spatial signals and theta frequency pacemaker inputs, they generate phase precessing action potentials that can coordinate theta sequences in place cell populations. We reveal novel constraints on sequence generation, predict cellular properties and neural dynamics that characterize sequence compression, identify circuit organization principles for high capacity sequential representation, and show that theta sequences can be used as substrates for association of conditioned stimuli with recent and upcoming events. Our results suggest mechanisms for flexible sequence compression that are suited to associative learning across an animal's lifespan.

**\*For correspondence:**
mattnolan@ed.ac.uk

[†]These authors contributed equally to this work

**Present address:** [‡]Gatsby Computational Neuroscience Unit, Sainsbury Wellcome Centre, University College London, London, United Kingdom

## Introduction

Whereas many behaviorally important events take place on timescales of seconds, neuronal membrane dynamics operate at a millisecond timescale. The discovery that during movement hippocampal place cells fire action potentials with timing that precesses relative to the hippocampal theta rhythm (*O'Keefe and Recce, 1993*), and that time-compressed representations of behavioral sequences occur as spike sequences within each theta cycle (*Skaggs et al., 1996*; *Dragoi and Buzsáki, 2006*; *Foster and Wilson, 2007*), suggests that hippocampal activity is organized so that computations on a millisecond neural timescale can address events on behavioral timescales. Thus, spike sequences within theta cycles may form a neuronal substrate for episodic and spatial memory (*Pastalkova et al., 2008*; *Lisman and Redish, 2009*; *Buzsáki and Moser, 2013*; *Wikenheiser and Redish, 2015*). Nevertheless, the circuit mechanisms through which theta sequences are generated are unclear and the mechanisms by which they may contribute to learning have received relatively little attention.

Several features of theta sequences that may be important for their computational functions pose challenges to models attempting to explain their generation through biophysically constrained mechanisms. First, the rate at which action potentials precess relative to the theta rhythm depends on an animal's speed of movement (*Geisler et al., 2007*). Second, phase precession occurs along arbitrary two-dimensional trajectories (*Huxter et al., 2008*; *Climer et al., 2013*; *Jeewajee et al., 2014*). Third, theta sequences emerge within theta waves that propagate along the dorsoventral

**eLife digest** Nerve cells in the brain exchange information via electrical impulses. In a given brain area, the electrical impulses at any particular moment can be thought of as forming a code that represents an aspect of the outside world. For example, groups of nerve cells in the hippocampus generate a type of code called a theta sequence, which represents a series of recent and upcoming events. The specific timing of electrical impulses within a theta sequence is crucial in creating certain types of memory.

There are two major classes of nerve cell in the brain: excitatory cells activate impulses in neighbouring cells, while inhibitory cells act to temporarily block impulses from other nerve cells. Many groups, or "circuits", of nerve cells contain combinations of both cell types to control how and when they communicate. Previous studies show that both types of cell are active within theta sequences, but it is not known precisely how they contribute to creating the sequences.

Chadwick et al. developed a new mathematical model that simulates how theta sequences can emerge from circuits of both excitatory and inhibitory nerve cells. The connections between these simulated cells are based on experimental data from real nerve cells in the hippocampus. The model predicts that inhibitory cells play an important role in generating theta sequences by interacting with groups of excitatory cells to coordinate the timing of electrical impulses. Furthermore, the model shows how memory capacity depends on these connections.

The next step following on from this work is to carry out experiments to test the model's predictions. This will include monitoring the same group of nerve cells in multiple different situations to find out how their theta sequences change, and recording electrical events in individual nerve cells during theta sequences. If the theory's predictions are confirmed this would lead to a deeper understanding of how our brains remember sequences of events.

axis of the hippocampus (*Lubenov and Siapas, 2009*; *Patel et al., 2012*). Fourth, while place cells across the dorsoventral axis have field sizes that vary over an order of magnitude, spike phase nevertheless usually advances by a maximum of a single theta cycle across their place field indicating that the rate of phase precession varies dorsoventrally (*Kjelstrup et al., 2008*). Finally, to successfully distinguish behavioral episodes, distinct theta sequences must be generated for experiences over an animal's lifetime, implying that sequence generation must be both flexible and have a high capacity (*Chadwick et al., 2015*). We previously introduced a phenomenological model which demonstrates that experimentally observed theta sequences can be accounted for by phase precession in independent place cells (*Chadwick et al., 2015*). This is in contrast to suggestions that synaptic coordination within and between cell assemblies is required to explain theta sequences (*Tsodyks et al., 1996*; *Harris et al., 2003*; *Harris, 2005*; *Maurer and McNaughton, 2007*; *Geisler et al., 2010*; *Lisman and Redish, 2009*; *Wikenheiser and Redish, 2015*; *Wang et al., 2015*). Several cellular mechanisms for independent phase precession have been proposed (*O'Keefe and Recce, 1993*; *Mehta et al., 2002*; *Harris et al., 2002*; *Lengyel et al., 2003*; *Burgess et al., 2007*; *Leung, 2011*; *Chance, 2012*), but none appear able to account for the challenges above while maintaining consistency with the hippocampal circuitry (see *Figure 1—source data 1* and Discussion). Thus, the biophysical mechanisms through which an independent phase coding scheme could be implemented within the CA1 circuitry while accounting for known computationally important properties of theta sequences are not clear.

The possible mechanisms underlying phase precession in CA1 are heavily constrained by the architecture of the CA1 circuit. CA1 pyramidal cells make few direct connections with one another (*Anderson et al., 2007*) (but see *Yang et al., 2014*), suggesting that phase precession in CA1 arises through some combination of intrinsic cellular properties, external inputs to the circuit and local interactions between pyramidal cells and interneurons. Major sources of input to CA1 include spatially modulated signals from CA3 and from the entorhinal cortex, and temporally patterned GABAergic inputs from the medial septum, which target hippocampal interneurons and act as a pacemaker to entrain theta oscillations in the circuit (*Freund and Antal, 1988*). Previously proposed mechanisms for independent phase precession focus on integration of signals by place cells

(*O'Keefe and Recce, 1993*; *Mehta et al., 2002*; *Harris et al., 2002*; *Lengyel et al., 2003*; *Burgess et al., 2007*; *Leung, 2011*; *Chance, 2012*). However, many interneurons also fire spikes that precess in phase against the theta rhythm, with interneuron phase precession exhibiting strong functional coupling to individual pyramidal cells (*Maurer et al., 2006*; *Geisler et al., 2007*; *Ego-Stengel and Wilson, 2007*). Thus, we asked whether phase precession underlying sequence generation could originate from interneuron dynamics. To address this possibility we introduce a minimal circuit model in which phase precession and theta sequences are generated through interactions between place cells and interneurons driven by pacemaker inputs. In contrast to the view that phase precession in interneurons is inherited synaptically from phase precessing place cell assemblies (*Maurer et al., 2006*; *Geisler et al., 2007*), interneuron phase precession in our model is crucial for the coordination of spike timing in place cells and for the generation of theta sequences. Due to the transient functional coupling between place cells and interneurons, phase precession occurs dynamically whenever a place cell is driven by external inputs. Consequently, phase precession and theta sequences are generated de novo within the network, and slow input sequences are automatically compressed into theta sequences in networks of interacting pyramidal cells and interneurons.

Our model suggests that CA1 can function as a flexible compressor of its inputs to maintain a representation of temporal order occurring on a behavioral timescale within a faster timescale suitable for synaptic processing in downstream brain areas. The model enables predictions of pacemaker dynamics which account for velocity-dependence of network activity and dorsoventral organization of sequence generation, and predicts network configurations that may underlie the dissociation of phase precession and theta sequences (*Feng et al., 2015*; *Middleton and McHugh, 2016*). The proposed mechanism not only generates sequences encoding spatial trajectories, but can also function as a general purpose circuit with a remarkably high capacity for encoding temporally extended sequences of events. We show how such a compression of ongoing experience into theta cycles enables flexible learning of behavioral associations through spike timing dependent plasticity (STDP). Thus, CA1 may compress ongoing experiences during theta states into fast neural activity patterns suitable for online learning and decision making.

## Results

### Phase precession emerges in coupled interneuron-pyramidal cell pairs

Since theta phase precession in independent neurons is sufficient to account for experimentally observed theta sequences (*Chadwick et al., 2015*), we first aimed to identify circuit mechanisms that account for experimentally observed features of phase precession in single neurons. Whereas in many previous models precession is assumed to arise from oscillatory drive targeting place cells, the frequency of theta is established by septal GABAergic projections to hippocampal interneurons (*Freund and Antal, 1988*), which in turn coordinate the spiking activity of local CA1 pyramidal cells (*Royer et al., 2012*). We therefore reasoned that phase precession could emerge from the dynamics of interneurons driven by pacemaker inputs and interacting with pyramidal cells. To explore this possibility we constructed a minimal network model containing a single interneuron and pyramidal cell, with synaptic connectivity based on this architecture (*Figure 1A*). The interneuron is driven to fire tonically by a constant depolarizing current, while pacemaker drive from the medial septum to the interneuron is simulated by an 8 Hz oscillatory current, which is sufficient to fully entrain spiking activity of the interneuron when the pyramidal cell is inactive (*Figure 1B–C*). In this case, output from the interneuron drives rhythmic subthreshold theta frequency inhibitory synaptic potentials in the pyramidal cell (*Figure 1B*). When spatial input to the pyramidal cell is simulated by a suprathreshold external drive, the resulting synaptic drive to the interneuron initiates phase precession in the coupled pair of cells, causing their firing frequency to elevate above that of the pacemaker theta and their firing phase to advance continuously over the place field (*Figure 1D–E*). When the pyramidal cell is transiently driven by slow depolarizing current, the phase of spikes fired by the interneuron and by the pyramidal cell advances through a full 360 degrees relative to the 8 Hz pacemaker input. Hence, whenever pyramidal cells are activated by slow depolarizing drives, the basic architecture of the CA1 circuit, along with pacing inputs from the medial septum, is sufficient to generate phase precession in pyramidal cells and interneurons.

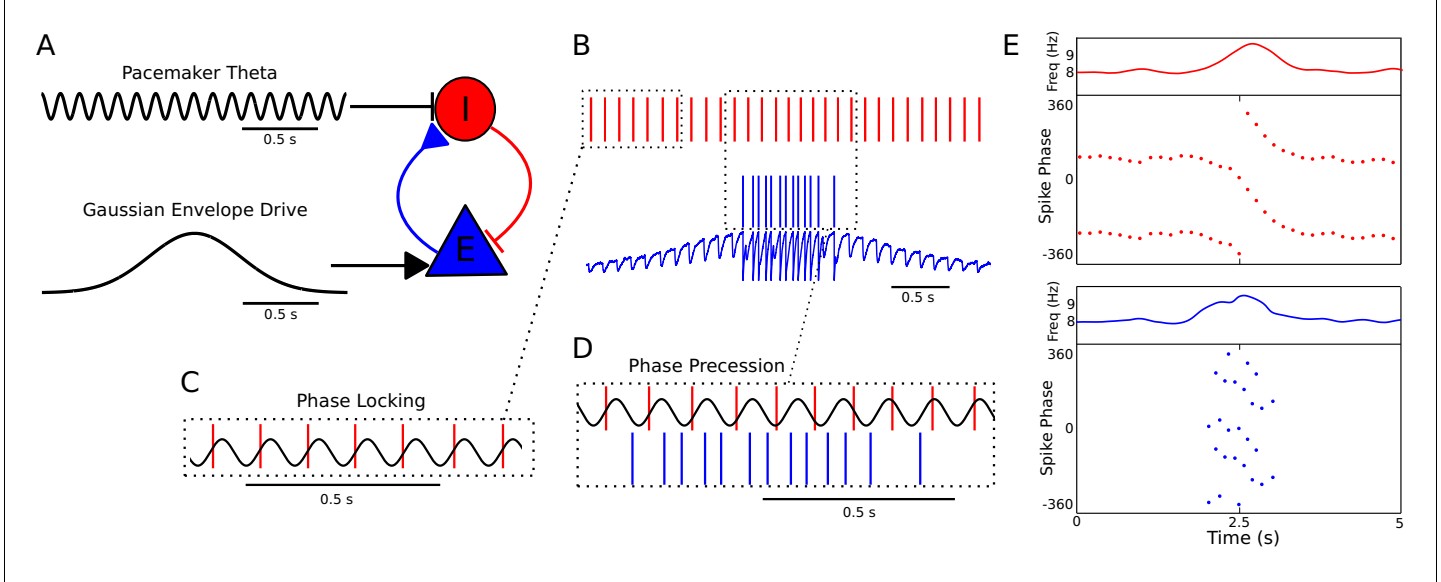

**Figure 1.** A minimal CA1 circuit model for theta phase precession. (**A**) An interneuron (red) is driven by a pacemaker theta oscillation from the medial septum. The interneuron synapses reciprocally onto a pyramidal cell (blue). The pyramidal cell is driven by slower external inputs occurring over behavioral timescales. (**B–E**) A simulation of this network as the animal crosses the place field of the pyramidal cell. (**B**) Interneuron spiking activity (red lines) and pyramidal cell spikes (blue lines) and membrane potential (blue trace). (**C**) A sample of the interneuron spike train when the pyramidal cell is inactive (i.e., outside of the place field), with the pacemaker rhythm overlaid for reference. In this case, the interneuron locks to the pacemaker input. (**D**) A sample of the interneuron and pyramidal cell spike trains inside the place field. In this case, the interneuron precesses in phase against the pacemaker input and the pyramidal cell fires in bursts which also precess in phase. (**E**) The membrane frequency in the theta band and the spike phases of the interneuron (red) and pyramidal cell (blue) corresponding to the data shown in parts (**A**)-(**D**). Phases are replicated over two cycles for clarity. Note that the pyramidal cell fires up to two spikes per theta cycle in this simulation.

The following source data is available for figure 1:

**Source data 1.** Table comparing the proposed model to previous models of phase precession.

To better understand the emergence of these phase precession dynamics, we developed a reduced model in which an interneuron is driven by weak pacemaker input and a slow depolarizing drive, and for which analytical solutions can be obtained (*Figure 2A*, see Materials and methods for details of model). During injection of a constant input current the model generates either stable phase locking or phase precession at a constant rate against the pacemaker drive ('frequency pulling') depending on the strength of depolarizing drive relative to the strength of pacemaker input (*Figure 2B,C*). Phase locking occurs for weak drives, where the interneuron becomes entrained to a fixed phase of the pacemaker input. Frequency pulling occurs for strong drives, in which case, for a given constant input current, the interneuron oscillates with a fixed frequency difference from the pacemaker, causing its phase to advance continuously relative to the pacemaker input. In the phase locking region, because the locking phase varies as a function of the input current, variation in the input current can be used to achieve variation through a maximum of 180 degrees of theta phases (−90 to 90 degrees, see *Figure 2B*). In contrast, in the frequency pulling region, phase precession continues indefinitely at a fixed rate for a constant input current.

Previous models that focus on place cells also generate phase precession by integration of theta drives and slow depolarizing drives, achieving variable phase locking through a range of 180 degrees (*Mehta et al., 2002*; *Harris et al., 2002*). However, the underlying dynamics and mechanisms in our model are distinct from these schemes in several ways. First, the excitation-dependent locking of spike phase observed in our model for weak depolarizing drives (the phase locking regime) is the result of the active entrainment of an ongoing intrinsic cellular rhythm to a pacemaker drive, rather than a passive summation and thresholding of inputs to a silent pyramidal cell as in previous models. Moreover, the frequency pulling regime in our model, in which the external drive

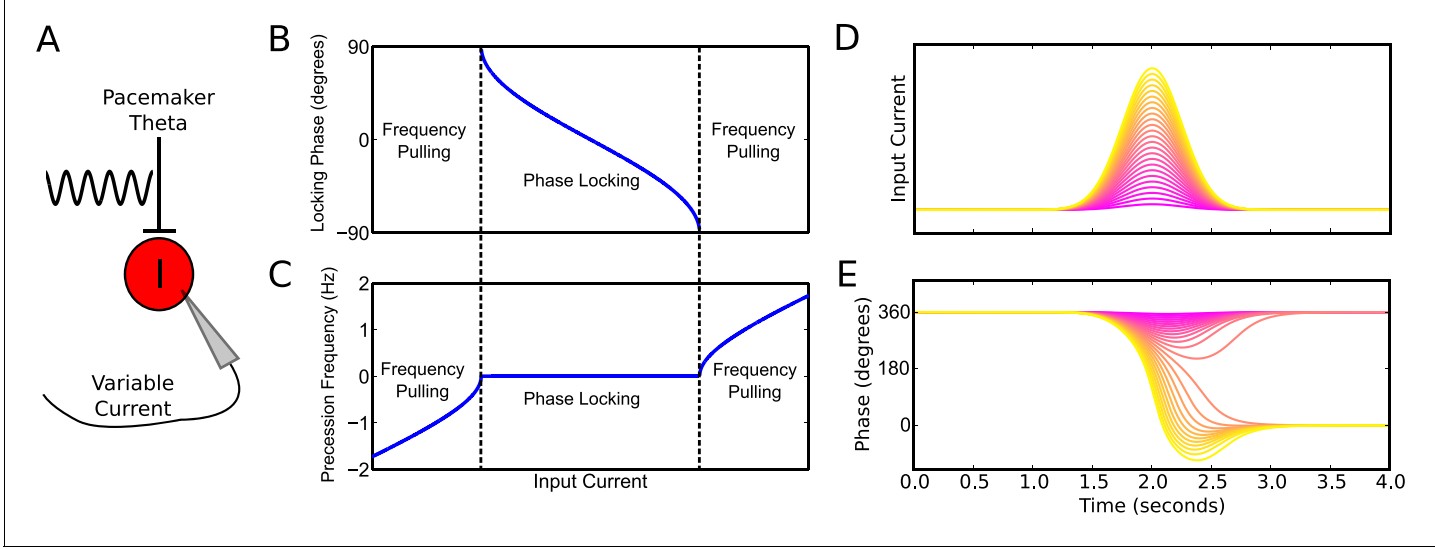

**Figure 2.** Phase precession and phase locking in a reduced model of an interneuron driven by depolarizing current and weak pacemaker drive. (A) Schematic of the model. (B–C) Steady state dynamics for a constant depolarizing drive, assuming a linear f-I curve. (B) Phase locking as a function of input current. (C) Precession frequency as a function of input current. For sufficiently strong currents, the interneuron oscillates with a frequency above that of the pacemaker (phase precession). For sufficiently weak currents, the interneuron oscillates more slowly than the pacemaker (phase regression). Note that for more biophysical f-I curves the phase regression regime may be absent. (D–E) Evolution of interneuron phase during a transient, slowly varying current injection. (D) Input currents with Gaussian profiles and a range of amplitudes. (E) Interneuron phase as a function of time, for each current profile shown in (D), showing a total of one cycle of phase precession for stronger drives and only transient phase precession before reversing in phase for weaker drives (purple).

The following figure supplement is available for figure 2:

**Figure supplement 1.** Phase precession is robust to the temporal profile of excitatory drive.

determines the *rate* of phase precession (*Figure 2C*), is not generated by previous models. The dynamics inside the place field in our model take place almost entirely within this novel frequency pulling regime, with the phase locking regime instead governing the dynamics outside of the place field and therefore the alignment of spike phase at place field entry. Because our model relies on the frequency pulling rather than the phase locking regime to produce phase precession, continuous phase precession can be generated for arbitrary input profiles of sufficient strength, and does not require a monotonically increasing ramp input as in previous models (*Figure 2—figure supplement 1*). Second, for symmetrical place fields previous schemes predict a phase advance towards the center of a place field, but a phase reversal as the input current is reduced on leaving the place field (*Melamed et al., 2004*). In contrast, when input currents are sufficient to drive the neuron into the frequency pulling domain in our model, then phase advances continuously throughout the input field (*Figure 2D,E*). Provided that inputs are sufficiently strong and sustained, the phase of interneuron firing advances through a full 360 degrees, with the rate of phase precession determined by the strength of the injected current (*Figure 2D,E*). Hence, this reduced model explains the dynamics observed in the network simulation of *Figure 1*. Specifically, the interneuron remains in a stable phase locking regime while the pyramidal cell is inactive, but enters the frequency pulling regime whenever the pyramidal cell provides sufficient synaptic input, producing phase precession. Phase precessing synaptic inputs from the interneuron coordinate the spike timing of the pyramidal cell and confer phase precession, but phase precession in the interneuron is relatively insensitive to the timing of pyramidal cell inputs, instead requiring only a sufficient increase in excitatory drive.

## Velocity-modulated precession frequencies are achievable through speed-dependence of synaptic currents

Experimentally the rate of phase precession in both place cells and interneurons increases with running speed, so that a constant relationship is maintained between spike phase and location (*Geisler et al., 2007*). Because phase precession in our reduced model depends on pacemaker amplitude and excitatory drive, the precession frequency can be flexibly modulated by varying either parameter without needing to adjust the frequency of the pacemaker oscillation (*Figure 2C*, Materials and methods). We therefore used the minimal circuit model of *Figure 1* to test whether variation of these inputs to the interneuron can account for the experimentally observed speed-dependence of phase precession in pyramidal cells and interneurons. The reduced model predicts that either a decrease in pacemaker amplitude or an increase in depolarizing drive to interneurons with running speed would generate an increase in the rate of phase precession with running speed. However, for stability the pacemaker amplitude must be small for low running speeds (see Materials and Methods). In this case the precession frequency can nevertheless be controlled independently through changes in the depolarizing drive with running speed. Indeed, we found that in the minimal circuit model a linear increase in pacemaker amplitude with running speed, combined with a linear increase in depolarizing current to interneurons with running speed, can generate an approximately linear increase in precession frequency while maintaining stable precession dynamics across running speeds (*Figure 3*). Hence, the dynamics required to maintain a fixed relationship between spike phase and place field position can be generated de novo in the local circuitry with inputs at a fixed theta frequency. Importantly, the predicted dependence on running speed of current input to the interneuron is consistent with findings of a velocity-dependent depolarizing current from glutamatergic circuits in the medial septum to interneurons in CA1 (*Fuhrmann et al., 2015*). Similarly, the predicted dependence of the pacemaker amplitude on running speed is consistent with the dependence on running speed of both the LFP theta amplitude in CA1 (*McFarland et al., 1975*;

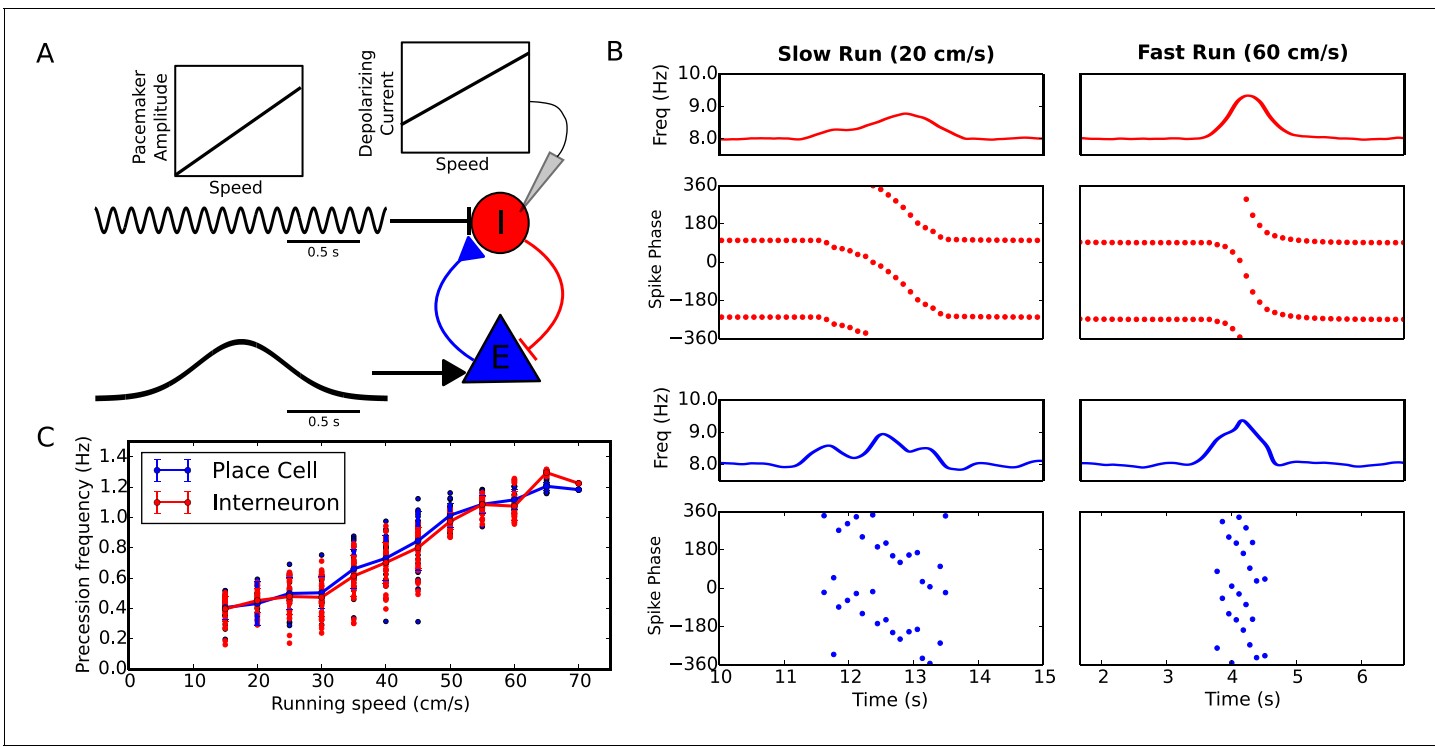

**Figure 3.** Running speed dependence of phase precession. (**A**) Illustration of the model circuit. To account for running speed dependence, pacemaker amplitude and depolarizing current amplitude are increased linearly with running speed. (**B**) Examples of phase precession at a slow and fast running speed, where the pacemaker amplitude and depolarizing current to interneurons are varied. (**C**) Phase precession frequency as a function of running speed. Individual dots illustrate the estimated precession frequency on a single lap.

*Maurer et al., 2005*; *Patel et al., 2012*) and the activity of inhibitory circuitry in the medial septum (*King et al., 1998*).

## Dorsoventral traveling waves and phase precession gradients can emerge from a common pacemaker drive

The phase of theta activity varies systematically across the dorsoventral axis of the hippocampus (*Lubenov and Siapas, 2009*), spanning a range of 180 degrees (*Patel et al., 2012*) and creating the appearance of a dorsoventral traveling wave. This variation is difficult to account for by temporal delays in a common pacemaker drive, which has led to the suggestion that entorhinal-hippocampal or intrahippocampal interactions are required to account for dorsoventral phase offsets (*Patel et al., 2012*; *Long et al., 2015*). We asked if the present model can account for these observations without the necessity for additional circuit components. In the reduced interneuron model, the range of phases that stably lock to pacemaker input is precisely 180 degrees, with locking phase depending on the strength of the excitatory current and pacemaker amplitude (*Figure 2B*). All other spike phases are unstable. This suggests that a gradient in excitatory inputs to interneurons (or alternatively a gradient in input resistance or some intrinsic membrane current) along the dorsoventral axis might be sufficient to generate the observed dorsoventral phase gradient, despite a coherent pacemaker input.

To test this hypothesis using more biologically plausible neuronal dynamics we simulated integrate and fire interneurons driven by the same pacemaker inputs, but different levels of depolarizing

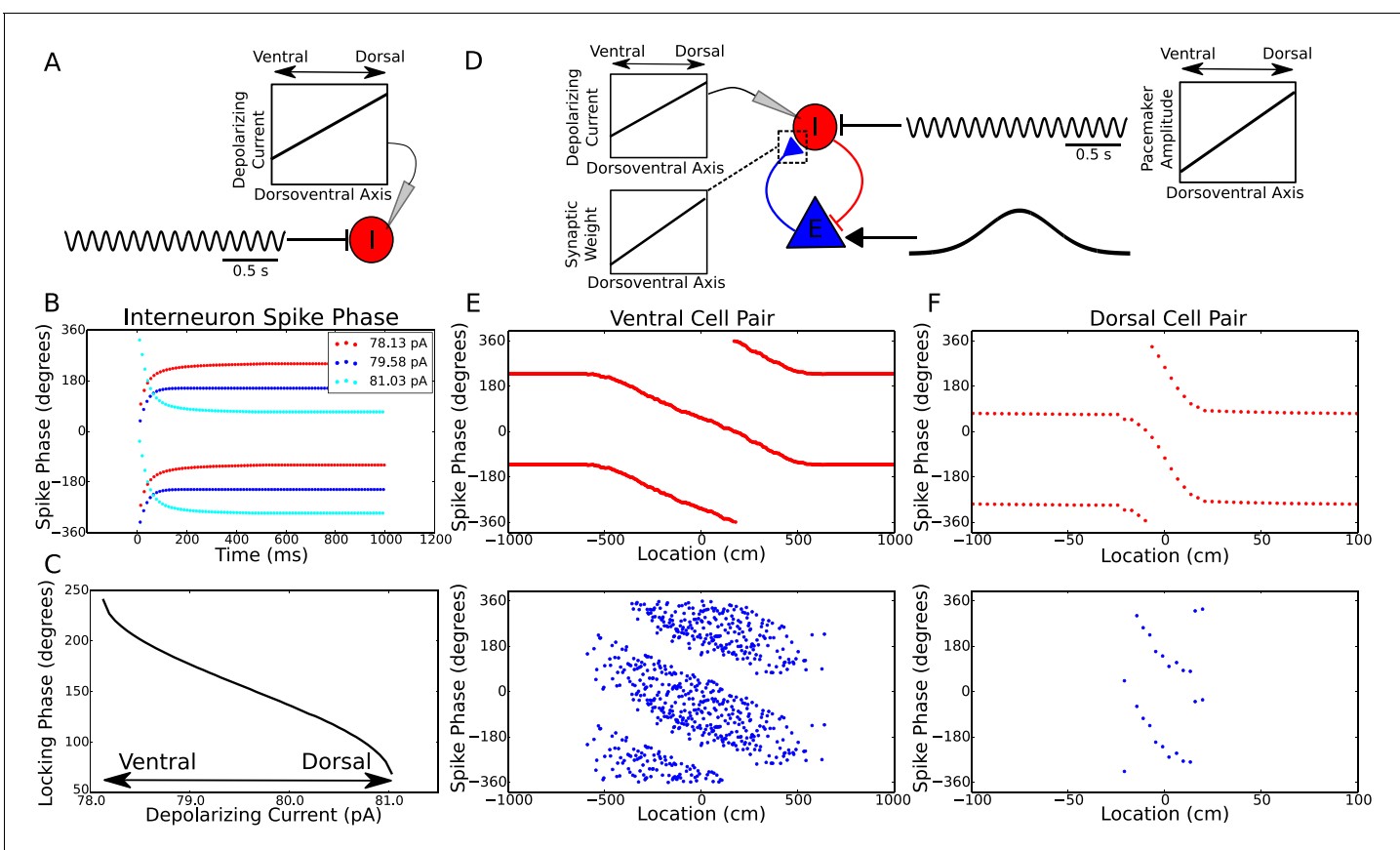

**Figure 4.** Theta dynamics across the dorsoventral axis. (**A**) Inputs to interneurons across the dorsoventral axis hypothesized to produce a gradient in theta phase. (**B**) Interneuron spike phases for three simulations with different depolarizing currents. (**C**) Interneuron locking phase vs depolarizing current (cf. *Figure 2B*). (**D**) A circuit model, and its dependence on dorsoventral location, which could produce simultaneous traveling theta waves and gradients in precession slope. (**E**) Phase precession in a ventral place cell/interneuron pair (place field size 10 meters). (**F**) Phase precession in a dorsal place cell/interneuron pair (place field size 0.3 meters). Note the change in both locking phase and precession slope from dorsal to ventral.

input currents (*Figure 4A–C*). This is equivalent to the full circuit model while the animal is outside of the place field and therefore the pyramidal cell is inactive. *Figure 4B* shows three examples of these simulations. In each case, the interneuron is attracted towards a stable locking phase of the pacemaker input, but the precise locking phase depends on the strength of depolarizing current. In *Figure 4C* we systematically analyzed how this locking phase depends on the strength of depolarizing current, finding a relationship remarkably similar to that predicted by the reduced model, including a range of 180 degrees of locking phases. Hence, in addition to explaining the change in precession frequency with running speed, the interplay between excitatory currents and pacemaker inputs can explain the phase gradient across the dorsoventral axis of the hippocampus, allowing the emergence of traveling theta waves based on variable locking to a single, common pacemaker input.

Place field size also varies along the dorsoventral axis of the hippocampus, ranging from less than one meter dorsally to approximately 10 meters ventrally (*Kjelstrup et al., 2008*). This gradient is associated with a concomitant gradient in the slope of phase precession, such that phase precesses through approximately one cycle both dorsally and ventrally (*Kjelstrup et al., 2008*). To test whether our minimal circuit model could account for these observations in addition to the traveling wave dynamics, we simulated place cell/interneuron pairs at the ventral and dorsal pole of CA1, with place field sizes of approximately 10 meters and 0.3 meters respectively, and interneuron locking phases separated by approximately 180 degrees (*Figure 4D–F*). We found that the gradient in both phase precession and theta phase along the dorsoventral axis could be accounted for simultaneously by a combination of a dorsoventral gradient in the amplitude of pacemaker drive, the depolarizing current to the interneuron and the strength of excitatory synaptic connections (*Figure 4E,F*). Thus, our proposed mechanism predicts that depolarizing current input to interneurons (or their excitability), the strength of excitatory synaptic connections from pyramidal cells to interneurons and the amplitude of the septal pacemaker drive all decrease from the dorsal pole to the ventral pole of the hippocampus (*Figure 4D*). In line with these predictions, theta power is observed to decrease from dorsal to ventral hippocampus (*Royer et al., 2010*).

## Robust phase precession is generated along two-dimensional trajectories

As a further test of the model we asked if in addition to accounting for phase precession on linear tracks, it can account for the properties of phase precession in open environments. In open environments, spikes always precess from late to early phases of theta, regardless of running direction (*Huxter et al., 2008*; *Climer et al., 2013*; *Jeewajee et al., 2014*). These dynamics arise naturally from the depolarizing current envelope in the present model if the animal passes in a straight line through the center of a place field at a constant speed (*Figure 1*). No additional inputs such as from head direction cells are required. Experimentally, a more complex feature of phase precession in open environments is observed on passes through the edge of the place field, in which case the firing phase advances through around 180 degrees before reversing through 180 degrees over the second half of the field (Supplementary Figure S2b in *Huxter et al., 2008*). In the present model, similar dynamics occur when the interneuron is not driven sufficiently strongly to pass through to the next cycle and is instead attracted back towards the initial phase (*Figure 2D–E* and *Figure 5A*). Our model is also consistent with sequences observed during backwards travel, in which theta sequences reflect the ordering at which locations are visited rather than heading direction (*Cei et al., 2014*; *Maurer et al., 2014*).

The phase advance and then reversal on passing through the edge of a place field results from failure of the weak synaptic depolarization to drive the model into the frequency pulling domain indicated in *Figure 2*. What happens when the synaptic drive is instead very large? We found that with strong and sustained inputs to place cells precession continues over multiple theta cycles (*Figure 5B*). However, the pacemaker drive to the interneuron confers robustness against this effect, as the interneuron can only precess through a discrete number of theta cycles and requires considerable additional input to precess through two cycles of theta rather than one. *Figure 5C* shows how the number of theta cycles precessed by the interneuron varies with the amplitude of the slow envelope. Over a broad range of input currents (*Figure 5C*), or more directly, a broad range of pyramidal cell spike counts (*Figure 5D,E*), the interneuron will precess exactly one cycle over the place field. For the choice of parameters used here, robust phase precession through one cycle in the

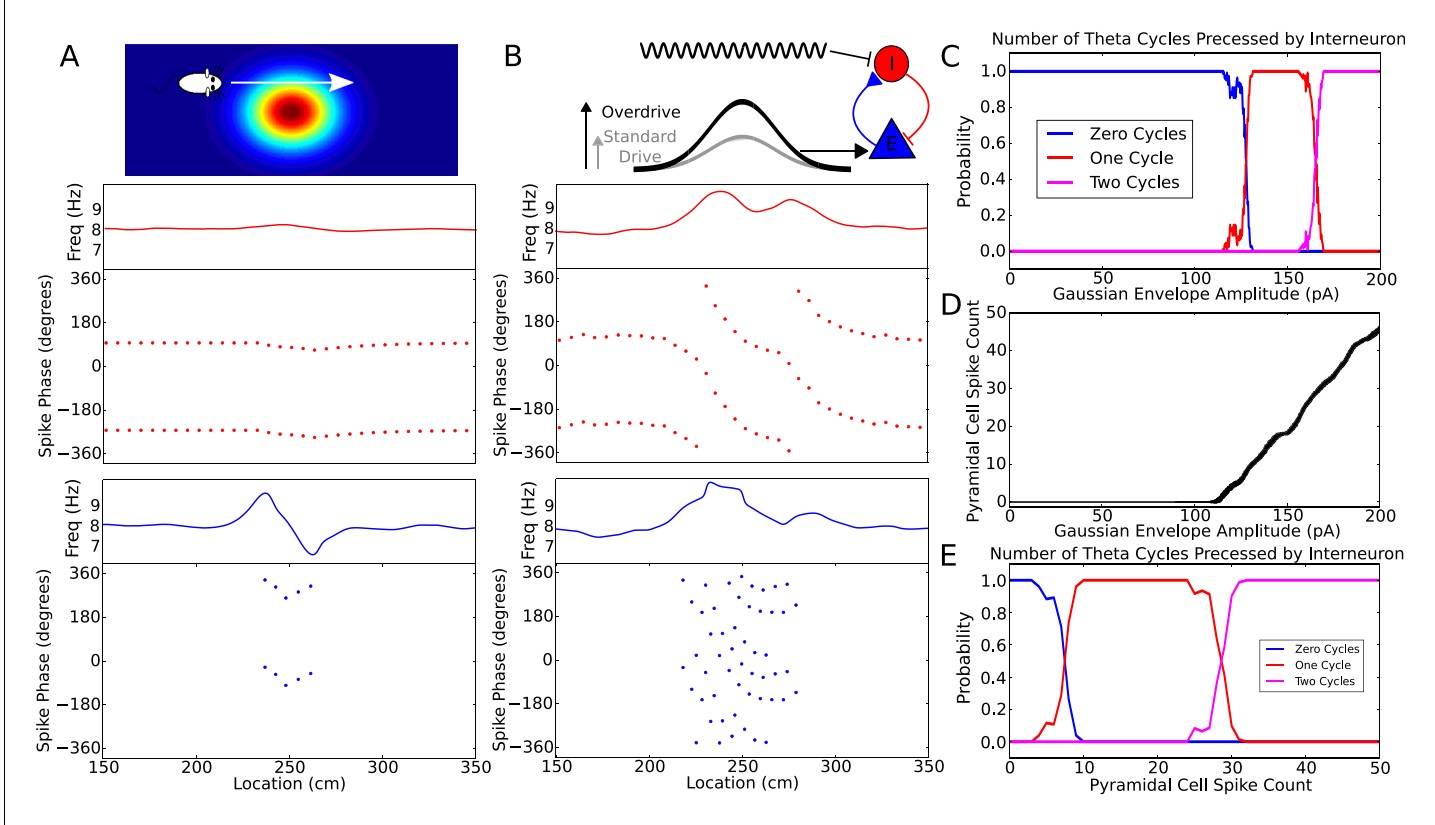

**Figure 5.** Robustness of phase precession to changes in the strength of place field drive. (**A**) Failure to precess through one full cycle. In this case, the external inputs were not strong enough to drive the interneuron past the threshold to be pulled into the next theta cycle, and instead it is pulled back towards the phase it started at. This is also seen in an initial increase followed by a decrease in frequency as the cell advances before reversing in phase against the pacemaker. (**B**) Precession through two full cycles. In this simulation, the amplitude of the slow envelope current was increased. This results in an increased firing rate of the pyramidal cell and hence an increased excitatory input to the interneuron. As a result, the interneuron received enough drive to pass through two cycles of pacemaker input. (**C**) The probability of an interneuron precessing through one, two, or three cycles of pacemaker theta phase as a function of the amplitude of the depolarizing envelope current onto the place cell. (**D**) The number of spikes fired by the place cell (with standard deviation shown as error bars) as a function of the amplitude of depolarizing envelope current. (**E**) The probability of the interneuron precessing through one, two, or three cycles of pacemaker theta phase replotted as a function of the number of spikes fired by the place cell.

The following figure supplements are available for figure 5:

**Figure supplement 1.** Phase precession is robust to transient intrahippocampal perturbation.

**Figure supplement 2.** Perturbation of spike phase during interneuron silencing.

interneuron occurs provided the place cell fires between 10 and 25 spikes in its place field. Thus, phase precession is sufficiently robust to allow considerable rate remapping, but the mechanism nevertheless places constraints on the coexistence of phase precession and rate remapping within place cells (*Allen et al., 2012*).

Theta phase precession has also been shown to exhibit considerable robustness against experimentally induced circuit perturbations. For example, when CA1 is transiently silenced (for ∼200 ms) and the theta rhythm is simultaneously reset, phase precession resumes in CA1 unperturbed upon recovery (*Zugaro et al., 2005*). We tested whether the model would exhibit similar robustness under such a perturbation by injecting a negative current into both place cells and interneurons to induce silencing while simultaneously resetting the phase of the external pacemaker drive. Indeed, we found that phase precession resumes upon recovery from this perturbation, as observed experimentally (*Figure 5—figure supplement 1*). Further robustness has been observed under optogenetic perturbations of the CA1 circuitry. Specifically, transient (1 s) silencing of somatostatin-positive

(SOM) interneurons has almost no effect on spike phase, altering mainly the burst firing of place cells, while silencing of parvalbumin-positive (PV) interneurons appears to introduce a small shift in average spike phase without compromising phase precession overall (*Royer et al., 2012*). We replicated this experimental protocol by injecting a 1 s negative current pulse into the interneuron as the animal crossed the place field. When analyzing the resulting data using the methods of Royer and colleagues, we found a shift in spike phase of a similar magnitude and direction to that reported in experimental data (*Figure 5—figure supplement 2A,B*). These findings can be explained as follows. The interneuron coordinates the pyramidal cell's theta activity until place field entry so that pyramidal cell spike phase is correctly aligned at the start of the place field. Upon interneuron silencing, the pyramidal cell's activity becomes independent of the theta rhythm, and depends only on the slow depolarizing drive. Nevertheless, because the pyramidal cell continues to spike tonically at a frequency higher than the theta rhythm during interneuron silencing, its spikes shift in phase continuously (i.e., precess) against the theta rhythm over the place field. This precession within the place field, combined with the phase alignment at place field entry provided by the interneuron before the onset of optogenetic silencing, generates the apparent phase shift of *Royer et al. (2012)* in the trial-averaged data. In contrast to this transient manipulation, we expect that phase precession would be severely disrupted in experiments where phase precessing interneurons are silenced over an entire lap, so that the phase at place field entry is not correctly aligned.

In summary, the model we outline here provides a robust mechanism for phase precession consistent with the circuitry in CA1. The model accounts for the key features of phase precession observed in CA1, including the dependence on running speed, place field size and dorsoventral location, phase precession along two-dimensional trajectories, the coupling of phase precession between place cells and interneurons, dorsoventral traveling theta waves and robustness to circuit perturbations.

## Efficient and flexible sequence compression depends on network configuration

While the model that we propose in *Figure 1* generates phase precession using only an isolated place cell and interneuron, CA1 place cells are embedded into much larger networks in which only 7–11% of neurons are interneurons (*Woodson et al., 1989*; *Aika et al., 1994*; *Bezaire and Soltesz, 2013*). The large disparity between the number of place cells and interneurons demands that a single interneuron in the model must couple to multiple pyramidal cells and generate phase precession in each one. To test if this is possible, we first simulated a single interneuron coupled synaptically to two pyramidal cells. We find that, when each pyramidal cell receives a depolarizing drive at a different time, the interneuron can be recruited for phase precession independently by each pyramidal cell (*Figure 6*). In this case, the interneuron shows two phase precession fields. As a direct consequence, and in contrast to the case in which there is just one active pyramidal cell per interneuron, the model predicts that outside of their suprathreshold firing fields place cells have subthreshold phase precession fields, characterized by transient increases in the frequency of their theta-modulated inhibitory input when the other place cell is active (*Figure 6B*). However, if the pyramidal cells have overlapping place fields, these dynamics may be disrupted. In this case, the stronger synaptic input to the interneuron from two active place cells may increase its precession frequency (*Figure 2C*), causing it to precess over multiple cycles (*Figure 5B*). Moreover, as synaptic output from the same interneuron coordinates the theta activity of both place cells, their spiking may become synchronized. Thus, a single interneuron can support phase precession by more than one place cell, but overlap between the firing fields of place cells coupled to the same interneuron may disrupt precession-based codes by shifting the phase of coding relative to position and by causing multiple cycles of phase precession within a single firing field.

Given this potential sensitivity of the circuit to overlap between place fields of cells connected to the same interneuron, it is unclear whether our proposed mechanism can be extended to large networks with realistic numbers of interneurons and pyramidal cells. To address this we quantify the performance of larger networks while varying the density of active place cells per interneuron, and the spatial arrangement of the firing fields of place cells connected to the same interneuron (*Figure 7A–B*). In these networks each pyramidal cell couples to only one interneuron, and these connections are bidirectional, so that each interneuron couples to multiple pyramidal cells (see Materials and methods). This can be viewed as a simplified description of the interactions underlying

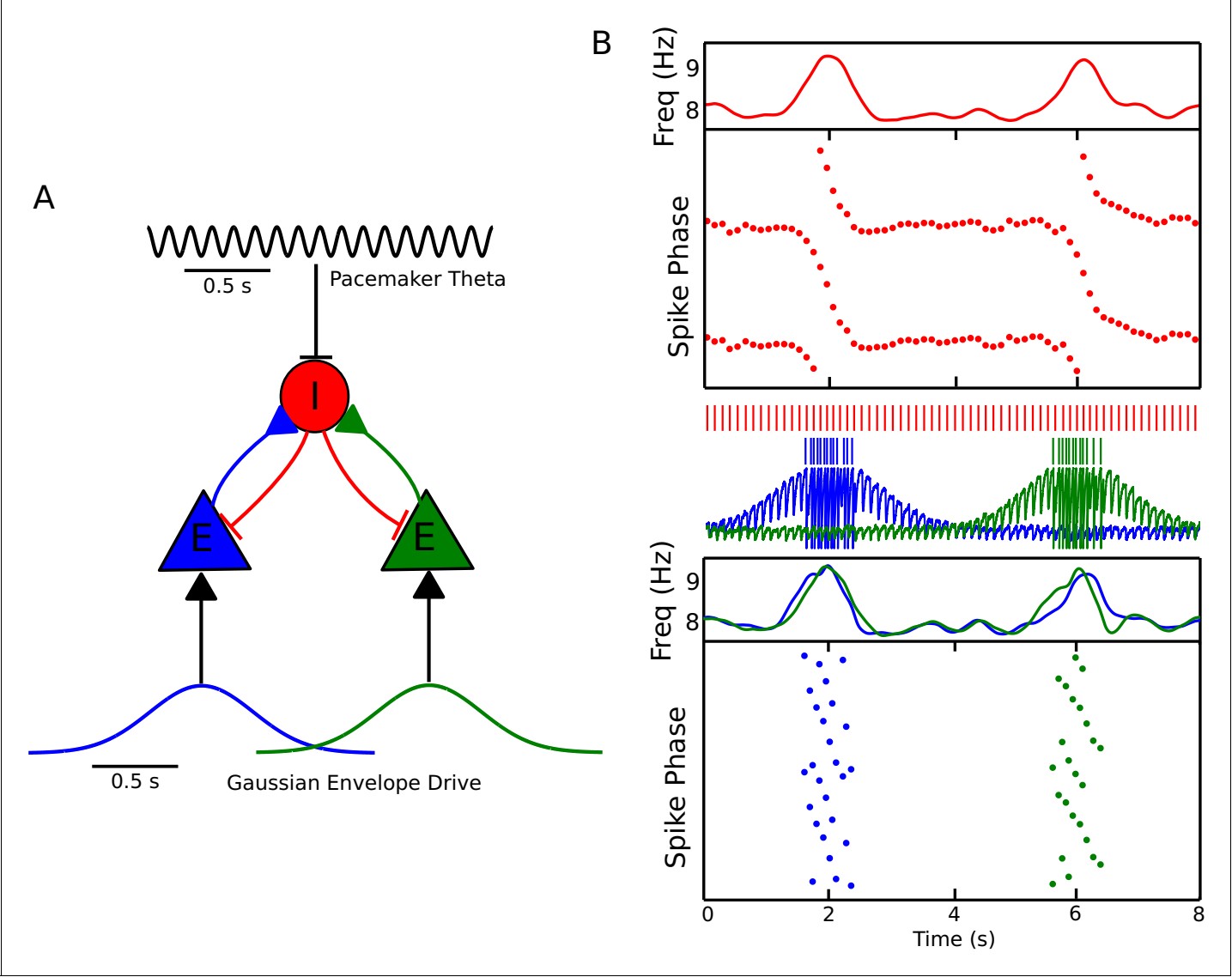

**Figure 6.** Recruitment of an interneuron for phase precession by multiple pyramidal cells. (A) Circuit diagram showing two pyramidal cells connected to the same interneuron and receiving slow envelope currents at different points in time. (B) Simulation of this circuit showing the intrinsic theta frequency, spike phases and membrane potentials. When the blue cell recruits the interneuron for phase precession as the animal crosses its place field, this is also reflected in phase precession of the membrane potential oscillation of the green cell while the animal is outside of its firing field (and vice versa).

phase precession, with other circuit interactions removed. These larger networks successfully compress slow input sequences into fast theta sequences when place field maps are sparse with low overlap (*Figure 7A* top). In contrast, such sequences do not emerge when place field maps are dense and have high overlap (*Figure 7A* bottom).

To more systematically quantify factors affecting sequence compression within the network, we introduced two distinct metrics which measure the extent to which spiking within theta cycles faithfully recapitulates the slow sequence of place field inputs. We call these the single-cycle theta sequence metric and the population phase precession metric (see Materials and methods for details). The single-cycle theta sequence metric measures the similarity between slow input sequences and individual theta sequences (*Figure 7C*, solid red line). The population phase precession metric measures the robustness and coherence of phase precession in a population of cells, and therefore serves as an averaged measure of sequence compression over a dataset (*Figure 7C*, solid blue line). For these metrics, correlations close to zero imply a lack of sequential organization and

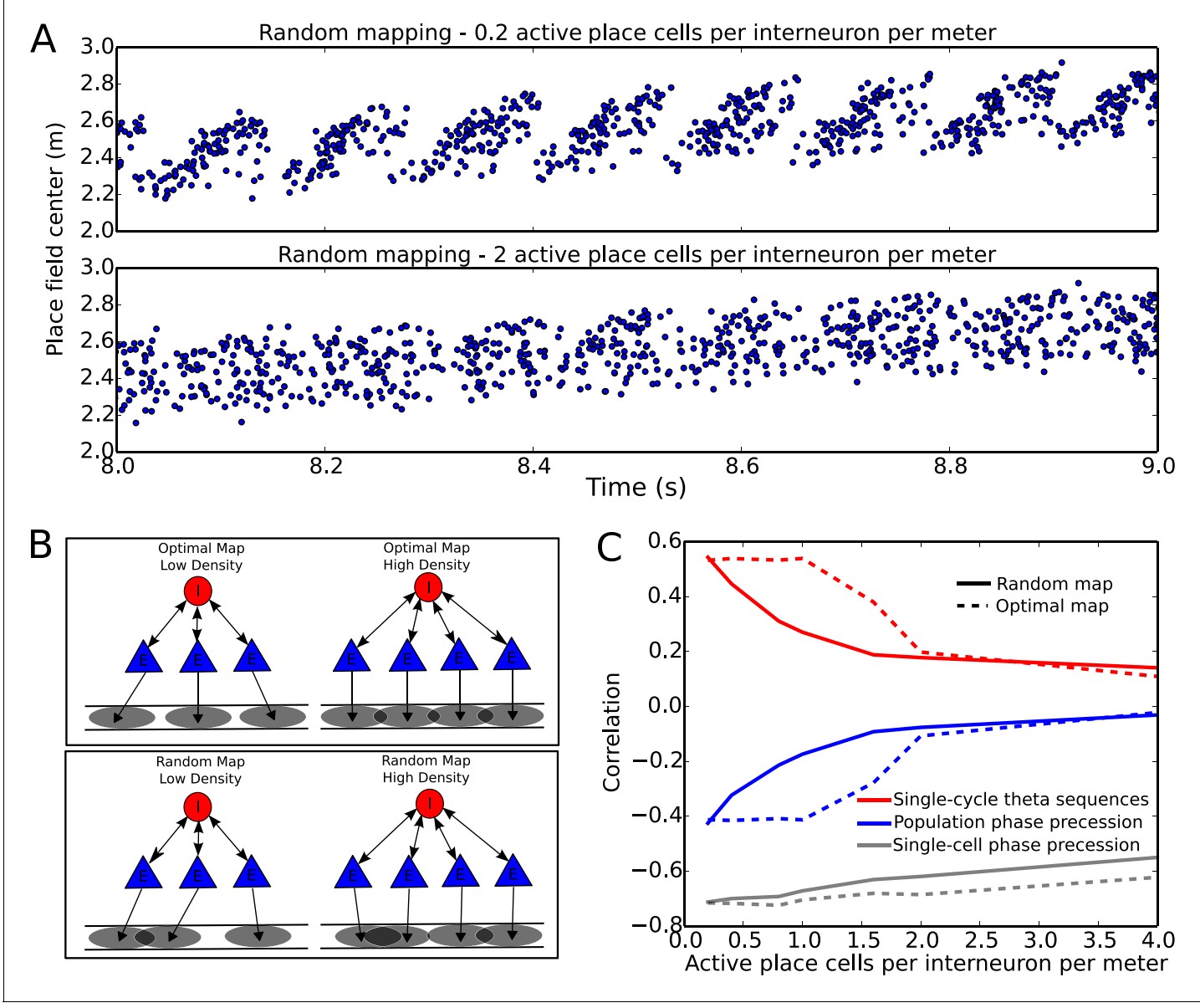

**Figure 7.** Compression of slow input sequences in CA1 networks. (**A**) Network simulations at low and high mapping densities. For sparse, random place field maps, input sequences are compressed into robust theta sequences. For dense, random place field maps, no such sequence compression is observed. (**B**) Top: Examples of optimal maps given two different place field densities. A set of place cells attached to the same interneuron are mapped onto a linear track. In an optimal map, their place field centers are organized such that their overlap is minimized. For a certain number of place cells per interneuron (here, four) overlap occurs even for an optimal map. Bottom: Example of random maps. The location of each place field on the track is drawn from a uniform probability distribution. In this case, a larger number of place cells per interneuron causes an increase in the probability that place fields will overlap. (**C**) Network performance vs number of active place cells per interneuron. As more place cells become active (or the number of interneurons is decreased), the compression of inputs into theta sequences is degraded (red and blue traces). This is caused by a drop in the coherence of phase precession in the population, despite a relatively constant phase-position correlation in individual place cells on single laps (blue trace vs gray trace).

The following figure supplements are available for figure 7:

**Figure supplement 1.** Three examples of random place field maps with a density of one active place cell per interneuron per meter, for which disruption of sequence compression occurs (*Figure 7C*).

**Figure supplement 2.** Robustness of phase precession under extraneous noise.

*Figure 7 continued on next page*

*Figure 7 continued*

**Figure supplement 3.** Putative mechanisms for removing disruption from network theta sequences.

**Figure supplement 4.** Distributions of single-cell phase precession strengths in random maps with varying degrees of disruptive place field overlap.

therefore poor performance, while strong (positive or negative) correlations signify the presence of sequential representations within theta cycles. Using these metrics, we tested how sequence compression varies depending on the properties of the place field map within the network. We observed decreases in the strength of both individual theta sequences and population phase precession with increasing place field density on the track (*Figure 7C*). Strikingly, this quantitative analysis also revealed that with random place field mappings network performance degrades continuously with increasing place field density, whereas with optimal place field mappings designed to minimize overlap in the place fields of cells coupled to the same interneuron (*Figure 7B*), high performance is maintained over a wider range of place field densities (*Figure 7C*, dashed red and blue lines).

In summary, both the number of active place cells in an environment and the spatial organization of their place fields influence the quality of sequence compression. In general, network performance is high when the spatial maps are sparse, but high levels of performance can also be maintained in denser spatial maps provided that the place fields of cells coupled to the same interneuron are well separated.

## Network reconfiguration can dissociate single-trial phase precession and theta sequences

Recent evidence suggests that theta sequences and phase precession on single laps may be dissociated under some circumstances, such that single-cell precession can occur without spatially ordered theta sequences. For example, on the first lap of a novel linear track place cells exhibit phase precession, in that their spikes advance continuously in phase against the theta rhythm, but the phase lags between cells are initially uncoordinated and do not generate population theta sequences until after further experience (*Feng et al., 2015*). Similarly, when input from CA3 is permanently absent, robust phase precession is observed in each cell while spatially organized theta sequences fail to emerge (*Middleton and McHugh, 2016*). To test whether such a dissociation of phase precession and theta sequences is consistent with our model, we asked whether the changes in sequence compression observed in the simulations of *Figure 7* are caused by changes in the robustness of phase precession in individual place cells, or whether they result from changes in the timing relationships between groups of place cells (i.e., a decoherence of phase precession in neuronal populations). When we quantified the fidelity of phase precession for individual cells on single laps (see Materials and methods) we found that robust single-unit phase precession persists as place field overlap is increased (*Figure 7C*, gray line), despite the disruption of single-cycle theta sequences and population phase precession. Thus, sequence disruption is caused by a decoherence of phase precession within the population. This decoherence is caused by indirect interactions amongst place cells with overlapping place fields and shared interneurons (e.g. see *Figure 7—figure supplement 1*).

Given this network configuration-dependent disruption of population activity in our model, we wondered if extraneous noise impacts phase precession and population sequences, and whether interneuron and pyramidal cell noise have similar or dissociable effects on circuit function. We found that with increasing amplitude noise injected into pyramidal cells, single-cell phase precession and population sequences were impaired in parallel (*Figure 7—figure supplement 2A–C*). This is distinct from increasing place field overlap, which disrupts population sequences but not single-cell phase precession. In contrast, increasing noise injected into interneurons disrupts population sequences while leaving phase precession intact at the single trial level, revealing an additional mechanism for dissociating phase precession from population sequences (*Figure 7—figure supplement 2D*). These results underscore the distinct roles of interneurons and pyramidal cells for generating phase precession and population sequences in the model.

We next sought to establish how the experience dependent reorganization of network activity observed by Feng and colleagues might occur. Our model suggests two potential mechanisms. First,

plasticity between place cells and interneurons could adjust synaptic weights such that place cells with overlapping place fields no longer couple strongly to the same interneurons (*Figure 7—figure supplement 3A*). Second, the slow envelope inputs to place cells could rapidly reorganize in order to minimize the overlap of place fields of cells coupled to the same interneurons (*Figure 7—figure supplement 3B*). If a plasticity mechanism were in place, synaptic changes which allow sequential activity in a new environment would cause disruption in previously stored maps. In contrast, place field reorganization could enable multiple stable maps to be formed without disruption or interference between different representations. Experimental evidence suggests that place field activity is indeed reorganized upon exposure to a novel environment, including a sparsification of the CA1 place code and a decrease in the number of active place cells (*Frank et al., 2004*; *Karlsson and Frank, 2008*). Whether such a reorganization mediates removal of unwanted place field overlap as we predict here is yet to be determined. Further evidence suggests that such a mechanism may depend crucially on plasticity in CA3 to CA1 connections (*Dragoi and Tonegawa, 2013*). Hence, permanent silencing of CA3 would be expected to disrupt CA1 theta sequences without affecting phase precession in our model, as observed by *Middleton and McHugh (2016)*.

## Theta sequences can be generated in a large number of spatial maps

What constraints does this sensitivity of sequence generation to connectivity impose on spatial mapping? Intuitively, as more place cells are connected to a given interneuron, the fraction of place cells that can be active in a given environment without interfering with sequence generation becomes smaller. This intuition can be formalized by adopting a simplified model in which place cells can map to different locations on a linear track, under the constraint that place cells which functionally couple to the same interneuron cannot map to locations within a certain distance of each other, which we termed the exclusion zone (see Materials and methods). With this model, we find the maximum fraction of pyramidal cells, $F$, which can express place fields in a given map is:

$$F < \frac{N_I}{N_P} \frac{L}{D} \tag{1}$$

where $N_I$, $N_P$ are the number of interneurons and pyramidal cells respectively, $L$ is the length of the track and $D$ is the size of the exclusion zone (approximately the size of a place field). The above inequality gives a bound on the density of the spatial representation. It implies that spatial maps generated by this network must be sparse and that the required sparsity depends on the ratio of pyramidal cells to interneurons, and on the size of the place fields. If the network is close to this upper bound, there will be a high density of subthreshold phase precession fields in place cells and interneurons will phase process over most of the environment. If instead the network is operating well below this upper bound, so that the representation is sparser than the minimum requirement, there will be only occasional interneuron and subthreshold phase precession fields. While subthreshold phase precession fields have not yet been investigated, the density of reported interneuron phase precession fields can be high (see Figure 2 of *Maurer et al., 2006*), suggesting that CA1 networks may operate close to this bound.

Does the non-overlap constraint limit the capacity of the network for the representation of distinct environments and contexts? When we quantify the capacity of the network under the non-overlap constraint (see Experimental Procedures), we find that the number of distinct spatial maps, cell assemblies and sequences that can be generated by the network are each considerably larger than the number of environments, events or behavioral episodes that an animal could encounter within its lifetime. For example, assuming a population of 10,000 pyramidal cells of which 20% are active in each map, 1000 interneurons, an exclusion zone between place fields of 1 meter, a linear track of length five meters and that place field locations can be distinguished with a spatial resolution of 10 cm (a conservative estimate), the number of spatial maps in which coherent theta sequences are generated is greater than $10^{5000}$. For the same population of cells, assuming each cell assembly consists of 100 pyramidal cells, there over $10^{500}$ possible cell assemblies, and assuming a phase sequence consists of 7 cell assemblies (*Lisman and Idiart, 1995*) there are over $10^{1500}$ possible sequences. Hence, despite the constraints imposed by the coupling between groups of pyramidal cells and interneurons, the capacity of the network to encode distinct environments, contexts and episodes can be considered to be unlimited from an ethological perspective.

The above analysis quantifies the number of maps under which all pyramidal cells exhibit robust phase precession. However, experimental data show a distribution of phase precession strengths in simultaneously recorded pyramidal cell populations in CA1 (e.g., *Skaggs et al., 1996*; *Schmidt et al., 2009*). We therefore asked whether randomly organized place field maps in our model might be sufficient to account for typical distributions of single-cell phase precession strengths in CA1 populations, despite disruptive place field overlap. In random maps, we found that pyramidal cells exhibit a broad range of phase-position correlations (*Figure 7—figure supplement 4*). Even for very dense random maps, in which phase precession is severely disrupted on average, a substantial proportion of the population continued to exhibit robust phase precession. Thus, even randomly organized place field maps may be sufficient to account for experimentally observed phase precession statistics in individual place cells.

In summary, we find that overlap between the place fields of pyramidal cells which functionally couple to the same interneuron can disrupt sequence compression in the network. The level of disruption increases with the number of active place cells per interneuron. For random place field mappings, maintaining coherent sequence compression requires that place field maps are sparse. By introducing mechanisms to organize place field maps in order to avoid interference, coherent sequence compression can be maintained with much larger numbers of active place cells. While such mechanisms reduce the number of spatial maps available to the network, we find that even under these constraints, there is a practically unlimited capacity for generating distinct spatial maps, cell assemblies and theta sequences in the network.

## Flexible compression of arbitrary input sequences allows learning through STDP

How might downstream neurons receiving synaptic input from place cells use compressed sequences for computation? A longstanding hypothesis is that sequence compression enables the association of events through spike timing dependent plasticity (STDP) (*Skaggs et al., 1996*). Because STDP acts on events correlated on a timescale of tens of milliseconds it is not well suited to directly associating behavioral events (*Levy and Steward, 1983*; *Markram et al., 1997*; *Magee and Johnston, 1997*; *Bi and Poo, 1998*), but it may act on compressed theta sequences representing several seconds of recent and upcoming experiences (*Figure 8A,B*). Theta sequence compression in conjunction with STDP has been suggested to lead to asymmetry in the firing fields of place cells receiving place cell input (*Mehta et al., 2002*), but the use of compressed event sequences as conditioned stimuli in classical associative learning has not been evaluated.

We consider a population of CA1 pyramidal cells performing sequence compression on its inputs and projecting to a downstream neuron which receives a second strong input encoding some particular outcome or event of behavioral relevance, termed the *unconditioned stimulus* (US) (*Figure 8B*). When the US occurs, the downstream cell signals that event by firing action potentials. Importantly, because behavioral events extending up to several seconds into the past are represented in an orderly fashion along the descending phase of the theta cycle and events occurring up to several seconds into the future are ordered along the ascending phase, sequence compression using theta oscillations generates an absolute temporal reference frame in neural time for past, present and future events in real time on which STDP can act (*Figure 8C*).

The absolute temporal reference frame provided by the theta cycle enables the timing of the downstream US-driven action potential to determine the association made. If these downstream action potentials lock to the trough of the theta rhythm, a standard STDP rule will cause inputs from place cells centered on locations before the place where the US was experienced to undergo an increase in synaptic strength (*Figure 8C*). This circuit therefore implements associative learning, forming an association between the conditioned and unconditioned stimuli. In contrast, if the downstream cell were to lock to a theta phase other than the trough, this would introduce a temporal shift to behavioral time lags at which potentiation and depression of synapses occurs. For example, a downstream neuron which fires at the peak of the theta oscillation will cause a *decrease* in synaptic strength from neurons representing past locations and an *increase* in synaptic strength from CA1 pyramidal cells representing the future locations (*Figure 8D*). Thus, sequence compression with theta oscillations allows locations, or events, occurring in the past or future to be flexibly and selectively associated with a particular outcome by varying the spike phase of the downstream cell. The

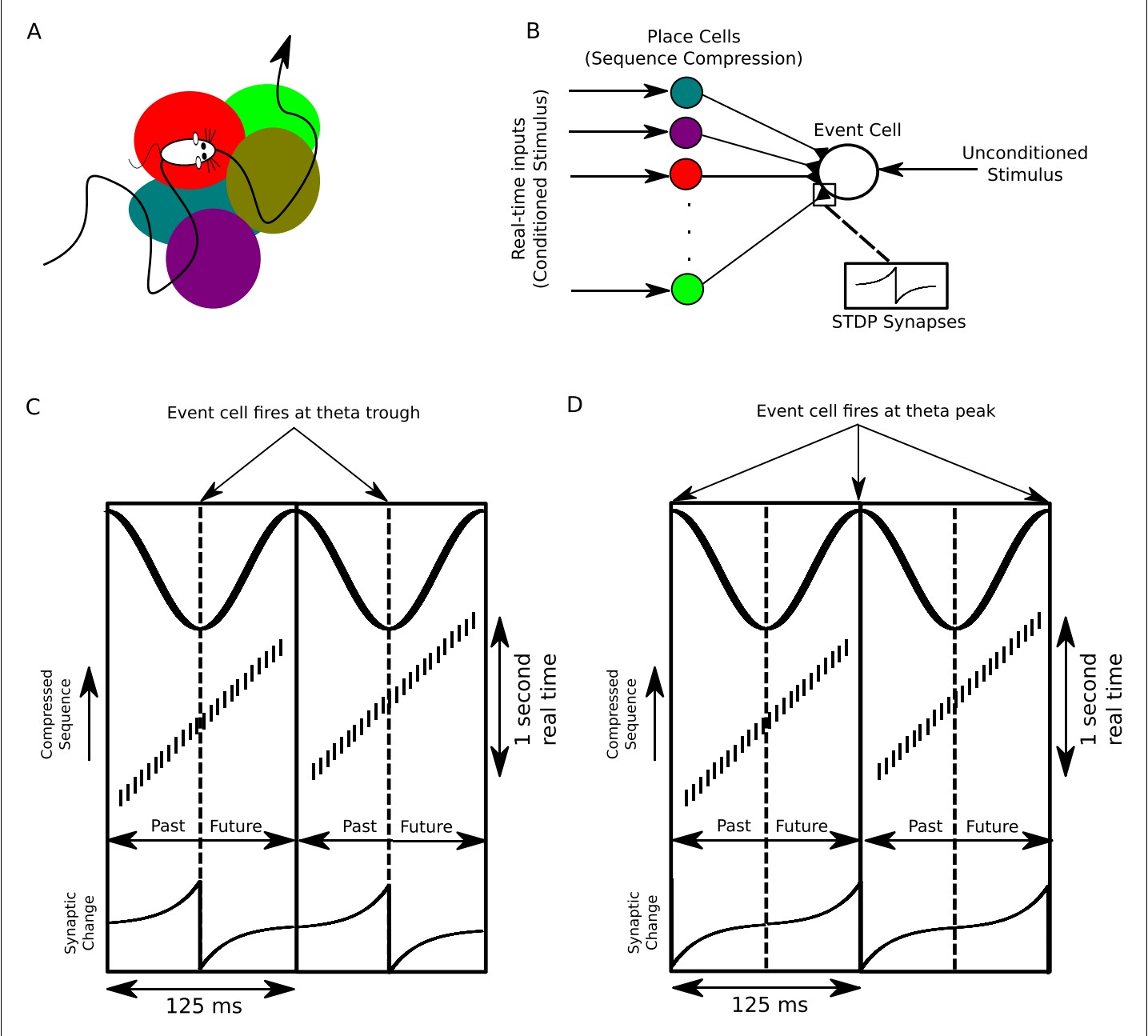

**Figure 8.** A proposed function of sequence compression for associative learning. (**A**) The animal explores an environment, activating different cells in CA1 in a particular temporal order on a behavioral timescale. (**B**) A population of CA1 place cells performs sequence compression on the slow Gaussian envelope inputs. These cells project onto a downstream neuron which signals some event of interest (the unconditioned stimulus). When this event occurs, this cell fires tonically at the trough of the theta cycle. Synapses from CA1 place cells to the event cell are modifiable via STDP. (**C**) During each cycle of the theta rhythm, CA1 cell assemblies representing past, present and future events in behavioral time are activated sequentially. At the trough of the theta cycle, place cells representing the animal's current location are active, whereas during the descending and ascending phases cells representing past and future locations respectively are active. If the downstream cell signaling the unconditioned stimulus fires an action potential at the trough of the theta cycle, STDP between pre- and post-synaptic spikes establishes an association between cells representing recently visited locations and the event. (**D**) If instead the downstream cell encoding spikes at the peak of the theta rhythm, an association between cells representing upcoming locations and this cell is formed, whereas cells representing recently visited locations and these cells have their synapses weakened (i.e., the temporal associations are reversed relative to those in **C**).

high capacity of the sequence compression mechanism that we propose here enables STDP to act on a practically unlimited set of potential behavioral experiences.

## Discussion

We show that compression of behavioral sequences into spike sequences on a timescale suitable for synaptic learning mechanisms can be achieved using a minimal network architecture based on CA1 and its inputs from the medial septum. This network architecture implements a novel mechanism for phase precession based on the dynamic integration of pacemaker and spatial signals by spontaneously active interneurons. Our model accounts for phase precession along arbitrary two-dimensional trajectories and across the dorsoventral axis of the hippocampus, and predicts tuning required to maintain phase precession with variation in running speed and spatial scale. It reveals that the phase precession of interneurons, previously assumed to be an epiphenomenon resulting from synaptic inputs from phase precessing place cell assemblies, may instead coordinate phase precession in pyramidal cells. A striking feature of this network is its large capacity for sequence generation. We also show how STDP mechanisms can implement classical conditioning of sequences encoding future or recent trajectories. Thus, the flexible sequence compression mechanism that we identify here may be a substrate for forms of associative learning that store relationships between conditioning stimuli and previous or future experiences. Our analysis indicates that sequence generation through single-cell phase precession would endow these learning mechanisms with sufficient capacity to operate across an animal's lifetime.

### Comparison to previous models for phase precession and theta sequence generation

In the model we propose here the dynamics of signal integration by interneurons are critical to phase precession and sequence generation. This is in contrast to previous models for phase precession which focus on place cells. A key difference is that interneurons generally exhibit ongoing rhythmic spiking activity throughout an environment, whereas place cells are typically silent across most of an environment, showing sustained firing activity only within spatially localized place fields. In our model phase precession requires entrainment of spontaneous spiking by a pacemaker input and acceleration of spiking due to further excitatory spatial input. The dynamics of our proposed model account for phase precession through a full 360 degrees, and suggest mechanisms for speed tuning and dorsoventral organization of phase that are consistent with experimental observations. We discuss below how the distinct dynamics of pyramidal cells lead to models of phase precession with different properties.

Previous models for phase precession face challenges in fully accounting for experimentally observed features of theta phase precession and sequence generation (see *Figure 1—source data 1*). An initial model for phase precession was based on interference between oscillations with different frequencies (*O'Keefe and Recce, 1993*). This model can account for phase precession observed through the full 360 degrees of a theta cycle. However, because it generates repeated spatial firing fields its predictions map more closely onto the properties of entorhinal grid cells than place cells in the hippocampus (*O'Keefe and Burgess, 2005*). Moreover, to account for phase precession in two dimensions interference models require heading modulated speed-dependent oscillatory signals, but experimental evidence for signals with the required properties is so far quite limited (*Harvey et al., 2009*; but see *Welday et al., 2011*). Other models rely on interactions between slow depolarizing inputs to place cells and oscillatory inputs to their soma and / or dendrites. These models generate a unidirectional phase advance over the place field using either an asymmetric ramp drive (*Mehta et al., 2002*; *Losonczy et al., 2010*; *Magee, 2001*) or using spike train adaptation so that firing ceases at the peak of a symmetric place field drive (*Harris et al., 2002*). However, these models appear able to achieve phase advances over the place field of only 180 degrees. Moreover, to avoid phase reversal in later parts of the firing field these models rely on sustained depolarization and elevated firing of place cells to maintain an advanced phase, whereas for many place cells phase continues to advance after the center of their firing field while firing rate and membrane potential depolarization drop (*Huxter et al., 2003*; *Harvey et al., 2009*). In contrast to these previous models, the network architecture we propose here is able to account for phase precession through 360 degrees, does not require sustained depolarization following the place field center, is compatible

with single rather than regularly repeating firing fields and does not rely on tuning of upstream velocity controlled oscillators.

Observations following experimental manipulations of phase precession also constrain models of the underlying circuit and cellular mechanisms. Intrahippocampal administration of cannabinoids disrupts the temporal organization of CA1 activity during theta cycles without altering firing rates (*Robbe and Buzsáki, 2009*), consistent with the scenario in our model when place field maps are relatively dense and unorganized (*Figure 7*) or when high levels of extraneous noise are injected into interneurons (*Figure 7—figure supplement 2*). Optogenetic inactivation of parvalbumin-positive interneurons (PV), but not somatostatin-positive interneurons (SOM), disrupts phase precession by shifting the firing phase of pyramidal cells towards the trough of the theta cycle (*Royer et al., 2012*), which is consistent with our model if phase precessing interneurons are generally PV-positive (*Figure 5—figure supplement 2*). Following a transient silencing of CA1 activity and resetting of the hippocampal theta rhythm, phase precession is relatively unperturbed (*Zugaro et al., 2005*), a finding which is also replicated by our model (*Figure 5—figure supplement 1*).

## Predictions for cellular and synaptic organization of CA1 circuits

Our proposed model for theta sequence compression makes a number of experimentally testable predictions. These can be grouped into core predictions of the model, and ancillary predictions that follow from constraining the model parameters to account for experimentally observed features of phase precession and to maximize the quality of sequence compression performed by the network. The core predictions are as follows. (1) Silent or inactive pyramidal cells should demonstrate subthreshold phase precession fields resulting from inhibitory input when their primary interneuron is activated by another place cell. This prediction should be testable through patch clamp recordings in awake animals (*Harvey et al., 2009*; *Epsztein et al., 2011*). (2) Groups of pyramidal cells which precess in tandem with a particular interneuron should have non-overlapping place fields, and if not will exhibit disrupted theta compression. This prediction may be testable with high density electrical recordings or advanced imaging methods. (3) Artificially depolarizing a place cell to generate a firing rate field should automatically produce phase precession. This requires that the interneuron driving phase precession is active, but otherwise should also be testable through awake patch-clamp recordings. (4) Phase precession in place cells should be accompanied by the presence of strong, phase precessing inhibitory synaptic inputs. (5) Entrainment of septal GABAergic inputs should set the basal theta frequency, but precession of CA1 interneurons and place cells against this basal theta should remain intact over a range of frequencies. (6) Phase precessing interneurons should show reciprocal synaptic connections onto pyramidal cells, and the interneuron to pyramidal cell synapse should be sufficiently strong to synchronize theta activity. (7) Sustained inactivation (e.g., over an entire lap) of phase precessing interneurons should abolish pyramidal cell phase precession.

Tuning of our model to account for experimentally observed features of phase precession leads to further predictions about expected properties of the network components. (1) Stable phase precession across different running speeds emerges when phase precessing interneurons receive a velocity-modulated excitatory drive and a pacemaker drive with velocity-dependent amplitude. (2) A dorsoventral gradient in excitation to phase precessing interneurons, alongside a gradient in pacemaker amplitude and excitatory synaptic strength, simultaneously generates dorsoventral traveling theta waves and changes in precession slope across the dorsoventral axis. We note that these ancillary predictions pertain only to the specific implementations that we have considered, and that alternative mechanisms are possible within the model circuitry. For example, dorsoventral traveling waves could equally emerge from a gradient in the phase of pacemaker input within our model, and alternative mechanisms for generating velocity-dependence of phase precession may also be possible within the circuit.

Because there is a greater number of pyramidal cells than interneurons in CA1, our model requires that each phase precessing interneuron couples to several pyramidal cells. For successful sequence generation pyramidal cells which couple to the same interneuron must have largely non-overlapping place fields. This constraint leads to predictions for network topographies that support and determine the quality of sequence compression. (1) For randomly organized place field maps, reducing the density of firing rate fields in the pyramidal cell population increases the quality of the sequence-compressed representation of behavioral events. (2) Sequence compression can be maintained with maximal performance at far greater place field densities when these place fields are organized so as to

minimize coactivity of pyramidal cells which precess with the same interneuron. Interestingly, this implies that CA1 does not exhibit topographically organized place field maps. This is consistent with an apparent lack of anatomical organization of place cells (*Redish et al., 2001*; *Dombeck et al., 2010*). (3) Because the maximum place field density is determined by the ratio of phase precessing interneurons to pyramidal cells, and because experimentally observed coding densities appear to increase along the dorsoventral axis, either: there are more phase precessing interneurons per pyramidal cell in the ventral hippocampus, the fraction of silent pyramidal cells in the ventral hippocampus is higher, or sequences are disrupted in the ventral hippocampus. When the constraint underlying these predictions is violated population theta sequences can be disrupted despite the presence of single-cell phase precession on individual laps. This may explain the dissociation of phase precession and theta sequences during exploration of novel environments (*Feng et al., 2015*) and when CA3 is silenced (*Middleton and McHugh, 2016*). In order to avoid such disruption of sequential activity, our model requires that CA1 networks can learn to decorrelate spatial maps following global remapping, most likely via a reorganization of firing rate fields in order to remove disruptive place field overlap. Understanding the learning rules and circuit mechanisms underlying such a decorrelation poses an interesting challenge, and points towards the importance of investigating the initial reorganization and stabilization of place field maps in novel environments (*Frank et al., 2004*; *Karlsson and Frank, 2008*), with a particular focus on the emergence of spatiotemporally structured representations within theta cycles (*Dragoi and Tonegawa, 2013*; *Feng et al., 2015*).

The model that we propose automatically compresses slow sequences of inputs occurring on timescales of seconds into fast sequences of spiking activity within each cycle of the network theta rhythm. This mechanism could in principle be implemented in parallel with some previously proposed mechanisms for phase precession. For example, in addition to inputs from local interneurons considered here, dendritic and somatic interference in pyramidal cells might also contribute to the phase advance over the first half of the place field (*Magee, 2001*; *Losonczy et al., 2010*), and oscillatory or phase precessing inputs from CA3 and entorhinal cortex might contribute to membrane phase in CA1 place cells (*Chance, 2012*; *Jaramillo et al., 2014*). Moreover, additional architectures could extend our proposed model. For example, in brain areas such as CA3 and entorhinal cortex, attractor mechanisms may generate the firing rate fields through local circuit interactions (*Samsonovich and McNaughton, 1997*), such that the circuit mechanism proposed here generates theta phase precession when excitatory neurons are driven by slow depolarizing inputs arising from within the local circuitry.

## Roles for sequence compression in memory functions of the hippocampus

How might theta sequences generated by pyramidal-interneuron interactions contribute to hippocampal-dependent learning and memory? Our analysis suggests a scenario in which, during theta oscillations, CA1 provides a time-compressed ongoing narrative of behavioral episodes (*Figures 1–7*). This enables downstream STDP mechanisms to form associations between ongoing behavioral events and specific outcomes such as reward or punishment (*Figure 8*). During sharp wave ripple events the CA1 network can then explore its state space and thereby test outcomes of different behavioral choices based on associations stored during theta activity (e.g., *Singer et al., 2013*; *Gomperts et al., 2015*). This allows a form of mental exploration in which possible behavioral sequences can be simulated and the likely outcomes determined based on associations learned during theta states (*Hopfield, 2010*). The model that we suggest here provides a mechanism for real-time generation of theta sequences with capacity for storing novel associations experienced across an animal's lifetime.

This proposed framework makes several additional experimentally testable predictions for neurons downstream from CA1. First, we predict that, during theta states, the spiking of downstream neurons encoding unconditioned stimuli is locked to the theta rhythm, but is not strongly influenced by the activation of specific cell assemblies in CA1. During sharp wave ripple events that take place following learning, we predict that activation of these same downstream neurons can be driven by cell assemblies in CA1 whose outputs were previously associated with the conditioned stimulus. In line with these predictions, reward responsive neurons in the VTA lock more strongly to the hippocampal theta rhythm than non-reward responsive neurons, and VTA neurons that lock more strongly to the hippocampal theta rhythm exhibit greater coordination with CA1 cell assemblies representing

reward locations during awake sharp wave ripple events (*Gomperts et al., 2015*). Second, to form associations between conditioned stimuli and rewards occurring in the future, present or past on behavioral timescales, the proposed learning mechanism predicts that the timing of downstream reward-encoding neurons relative to the theta rhythm should shift, firing near the peak for future rewards and near the trough when rewards have been obtained. This behavior has been observed in reward-encoding neurons in the ventral striatum, which precess in phase relative to the hippocampal theta rhythm as the animal approaches a reward site (*van der Meer and Redish, 2011*). Hence, during theta states a primary function of interneurons in CA1 and other hippocampal structures may be to support compression of ongoing events into neuronal sequences in order to store associations in synaptic projections to downstream brain areas, which may then be utilized during sharp wave ripple events for mental exploration, planning and decision making.

## Materials and methods

### Reduced model of interneuron phase dynamics

To understand how phase precession emerges in the circuit of *Figure 1*, we developed a reduced model of an isolated interneuron driven by a constant excitatory current and a pacemaker current (*Figure 2A*). In this simplified description, we treat the interneuron as an oscillator whose baseline frequency $\omega(I)$ is determined by the amplitude of the depolarizing current I through its f-I curve (note that we do not make any explicit assumptions about the form of this f-I curve in the reduced model, although in *Figure 2* we assumed a linear f-I curve). We consider the pacemaker input as a weak perturbation to this oscillator, which allows a reduced description of the interneuron in which only its phase is considered (e.g., *Ermentrout, 1986*). We show below that under some general assumptions the following equation is obtained (*Adler, 1946*):

$$\frac{d\Delta\phi(t)}{dt} = \Delta\omega - A\sin(\Delta\phi(t)) \tag{2}$$

where: $\Delta\phi = \phi - \theta$ is the instantaneous phase difference between the interneuron and the pacemaker input; the detuning $\Delta\omega(I) = \omega(I) - \omega_\theta$ is the frequency difference between the pacemaker input and the intrinsic frequency of the interneuron in the absence of pacemaker input; $A$ is the synchronization factor, which depends on the amplitude of pacemaker input and on the intrinsic properties of the interneuron (we describe this dependence in the derivation below). This equation approximates the phase relationship of the interneuron to the pacemaker input for different strengths of pacemaker drive and excitatory drive.

For a constant input current, *Equation (2)* generates two distinct dynamical states depending on the relative values of $A$ and $\Delta\omega$ (i.e., depending on the amplitude of pacemaker input and excitatory input to the interneuron). The first is stable phase locking and the second is phase precession. Phase locking occurs when $|\Delta\omega/A| < 1$, with a stable locking phase of $\Delta\phi_{\text{lock}} = \arcsin\left(\frac{\Delta\omega}{A}\right)$ (*Figure 2B*). Phase precession occurs when $|\Delta\omega/A| > 1$, where there are no stable phases and the interneuron precesses continuously, but nonlinearly, in phase against the pacemaker input (*Figure 2C*).

To derive *Equation (2)* and the dynamics described above, we assume that the pacemaker input is weak so that we may introduce an approximation based on its infinitesimal phase response curve $z(\phi)$. Specifically, the dynamics of an oscillator with frequency $\omega$ driven weakly by an external perturbation $Q(t)$ can be approximated by the reduced phase model:

$$\frac{d\phi}{dt} = \omega + z(\phi)Q(t) \tag{3}$$

where amplitude variations have been neglected. To model the case of an oscillator driven by a weak pacemaker (i.e., an interneuron driven by the septal theta rhythm) we consider a perturbation of the form $Q(t) = Q_0\cos(\theta(t))$. *Equation (3)* can then be expressed as:

$$\frac{d\Delta\phi}{dt} = \Delta\omega + z(\theta + \Delta\phi)Q_0\cos(\theta(t)) \tag{4}$$

where $\Delta\phi = \phi - \theta$ and $\Delta\omega = \omega - \omega_\theta$. If $z$ is also sinusoidal, the above equation can be further

approximated by averaging out fast fluctuations to obtain *Equation (2)*. To see this, we define the theta-average of a variable $X$ as:

$$\langle X \rangle_\theta = \frac{1}{2\pi} \int_0^{2\pi} X d\theta \qquad (5)$$

Averaging out fluctuations on a sub-theta cycle timescale then gives:

$$\left\langle \frac{d\Delta\phi}{dt} \right\rangle_\theta = \Delta\omega + Q_0 \langle z(\theta + \Delta\phi)\cos(\theta) \rangle_\theta \qquad (6)$$

which for sinusoidal phase response curves of the form $z(\phi) = z_0 - z_1 \sin(\phi)$ is:

$$\left\langle \frac{d\Delta\phi}{dt} \right\rangle_\theta = \Delta\omega + Q_0 \langle \cos(\theta)(z_0 - z_1 \sin(\theta + \Delta\phi)) \rangle_\theta \qquad (7)$$

$$= \Delta\omega - Q_0 z_1 \langle \cos(\theta)\sin(\theta + \Delta\phi) \rangle_\theta \qquad (8)$$

$$= \Delta\omega - \frac{1}{2}Q_0 z_1 \langle \sin(2\theta + \Delta\phi) + \sin(\Delta\phi) \rangle_\theta \qquad (9)$$

$$\approx \Delta\omega - \frac{1}{2}Q_0 z_1 \sin(\Delta\phi) \qquad (10)$$

where in the last line it was assumed that $\Delta\phi$ does not change over a single theta cycle. This recovers *Equation (2)* and provides an explicit formula for the synchronization factor in terms of the phase response curve and pacemaker drive $A = Q_0 z_1 / 2$. In other words, the synchronization factor depends on the amplitude of pacemaker drive and the sinusoidal component of the phase response curve of the interneuron.

The general solution to *Equation (2)* is given by:

$$\Delta\phi(t) = 2\arctan\left[ \frac{A - \sqrt{(\Delta\omega)^2 - A^2}\tan\left(\frac{1}{2}\sqrt{(\Delta\omega)^2 - A^2}(c - t)\right)}{\Delta\omega} \right] \qquad (11)$$

where $c$ is a constant determined by the initial conditions. This equation is valid for both the phase locking and phase precession regimes. In the case of phase precession, where $|\Delta\omega/A| > 1$, this gives the following precession frequency:

$$f = \sqrt{(\Delta\omega)^2 - A^2}/2\pi \qquad (12)$$

which is obtained by noting that $\tan$ is a $\pi$-periodic function, while $\arctan$ is a monotonic function. Assuming that the phase precession frequency scales with running speed $v$ and field size as $f = v/(2R)$, where $R$ is the radius of the place field (*Chadwick et al., 2015*), we obtain a constraint on the detuning and synchronization factor:

$$(\Delta\omega)^2 = A^2 + (\pi v/R)^2 \qquad (13)$$

which shows how the phase precession frequency can be controlled across running speeds and dorsoventral locations by changing the strength of pacemaker input and excitatory drive to interneurons.

For the stable phase locking case, where $|\Delta\omega/A| < 1$, the expression for the precession frequency yields complex values. To recover the steady state locking dynamics shown in *Figure 2B*, note that for complex arguments the $\tan$ function in *Equation (11)* becomes a $\tanh$, and in the limit $t \to \infty$ this $\tanh$ term tends to 1 so that $\Delta\phi(t)$ becomes independent of $t$ and the initial condition $c$. The rate at which the decay to steady state occurs will therefore vary with $|\Delta\omega/A|$.

## Numerical simulations
Simulations were performed using the Brian simulator (*Goodman and Brette, 2009*).

## Neuron model

We modeled the network using leaky integrate and fire neurons with conductance based synapses. For example, excitatory neurons were modeled by the following equation:

$$\frac{dV_m(t)}{dt} = -(V_m(t) - E_0)/\tau_m - \sum_{j=1}^{N_I} g_{I,j}(t)(V_m(t) - E_I)/C_m + I_{\text{Ext}}/C_m + \sigma_n \eta(t)/\sqrt{\tau_m} \tag{14}$$

where $V_m$ is the membrane potential, $E_0$ is the resting potential, $\tau_m$ is the membrane time constant, $g_{I,j}$ is the conductance of the synapse from presynaptic interneuron $j$, $E_I$ is the inhibitory reversal potential, $I_{\text{Ext}}$ is an external current input, $\sigma_n$ is the noise amplitude and $\eta$ is a random variable drawn from a standard normal distribution at each timestep. When the membrane potential $V_m$ reaches the threshold $V_\theta$, a spike occurs and the membrane potential is reset to $V_r$.

The synaptic conductances were exponentially decaying and governed by:

$$\frac{dg_{I,j}(t)}{dt} = -g_{I,j}/\tau_I + \sum_i w_j \delta\left(t - t_i^{(j)}\right) \tag{15}$$

where $t_i^{(j)}$ is $i$th spike of cell $j$ and $w_j$ the synaptic weight. Inhibitory neurons were modeled in the same way, but receiving excitatory rather than inhibitory synaptic conductances.

## External inputs

For interneurons, the external current was of the form:

$$I_{\text{Ext}}^I = I_0^I - I_\theta \cos(\omega_\theta t) \tag{16}$$

where $I_\theta$ is the amplitude of the pacemaker current. To simulate a trajectory through a place field, the external current injected into the place cell was of the form:

$$I_{\text{Ext}}^E(t) = I^E \exp\left(-\frac{|\mathbf{x}(t) - \mathbf{x_c}|^2}{2\sigma^2}\right) \tag{17}$$

where $\mathbf{x}(t) = \mathbf{x_0} + \mathbf{v}t$ is the trajectory of the animal through the place field. For simulations through the edge of the place field (*Figure 5A*) the trajectory was a straight line offset from the place field center $\mathbf{x_c}$ by 14 cm, otherwise the trajectory passed through the center.

## Synaptic connectivity

To simulate large scale networks in *Figure 7*, synaptic weights from pyramidal cell $i$ to interneuron $j$ were defined as:

$$w_{ij}^E = C_{ij} w^E \tag{18}$$

where $C_{ij}$ is a connectivity matrix with binary entries $C_{ij} \in \{0,1\}$ and $w^E$ is the strength of excitatory synapses in the network. The connectivity matrix satisfies the following conditions: $\sum_j C_{ij} = 1$ for all $i$ and $\sum_i C_{ij} = N_p/N_I$ for all $j$. This ensures that each pyramidal cell connects to exactly one interneuron and that each interneuron receives connections from the same number of pyramidal cells. Similarly, synaptic weights from interneuron $j$ to pyramidal cell $i$ were defined as:

$$w_{ji} = C_{ij} w^I \tag{19}$$

where the inhibitory connectivity matrix is simply the transpose of the excitatory connectivity matrix. This ensures that interneurons project to the same pyramidal cells from which they receive connections. There were no synaptic connections between neurons of the same type.

## Model parameters

The following parameters were fixed, independent of running speed:

$\tau_m^I = 40$ ms, $\tau_m^E = 20$ ms, $V_\theta = -50$ mV, $V_r = -70$ mV, $V_0 = -65$ mV, $C_m^I = 200$ pF, $C_m^E$ = 155 pF, $\tau_I = 10$ ms, $\tau_E = 2$ ms, $E_I = -70$ mV, $E_E = 0$ mV, $f_\theta = 8$ Hz, $w^E = 0.5$ nS, $w^I = 25$ nS, $\sigma = 40$ cm and a simulation time step of 0.1 ms.

### Running speed dependence

We varied several parameters to model the changes with running speed. First, the injected current into the place cell depends on running speed according to *Equation (17)*, where the width of the Gaussian in time varies with running speed. In addition to the temporal duration of current injection (as determined by the trajectory $\mathrm{x}(t)$), the amplitude of current input $I^E$ was varied with running speed. In addition, the noise to the place cell was varied with running speed. The amplitude and noise term were varied so that the width of the place field (measured as the distance from first to last spike on a single lap) and the number of spikes fired during a pass through a place field were constant across running speeds. Increasing noise tends to spread out the place field, whereas increasing the input current amplitude tends to increase both the number of spikes fired and the width.

The inputs to the interneuron were also running speed dependent. Specifically, the pacemaker amplitude $I_\theta$ and the baseline current $I_0^I$ were varied with running speed. By lowering the pacemaker amplitude, the range of input currents over which phase locking occurs is reduced, but the nonlinear transition from phase locking to phase precession is less severe. This effectively allows a wider range of currents over which slow phase precession can be achieved and increases the stability of phase precession within this range. For these reasons, we reduced the pacemaker amplitude at lower running speeds and also reduced the baseline current so as to allow a slow precession frequency.

The depolarizing current input (in pA) to the interneuron as a function of velocity (in cm/s) was set as:

$$I_0^I = 79.5 + 0.027v \tag{20}$$

The septal pacemaker input was:

$$I_\theta = 0.065v \tag{21}$$

The amplitude of current injection into the place cell was:

$$I^E = 110 + 0.5v \tag{22}$$

The noise to the place cell (in mV) was varied with running speed as:

$$\sigma_n^E = 1.75 - 0.025v \tag{23}$$

## Simulation of transient intrahippocampal perturbation

To model to experimental protocol of *Zugaro et al. (2005)*, we repeated the simulation with parameters as described above (with running speed $v = 40$ cm/s) and in addition injected a negative current of amplitude 50 pA into the pyramidal cell and 20 pA to the interneuron for a duration of 200 ms, centered on the peak of the place field input, while simultaneously resetting the phase of the pacemaker drive. These parameters were sufficient to generate silencing for around 200–250 ms in both cells.

## Simulation of interneuron silencing experiment

To model the experimental protocol of *Royer et al. (2012)*, we again repeated the simulation with parameters as described above ($v = 40$ cm/s), but in this case delivered a negative current of amplitude 10 pA to the interneuron for a duration of 1 s. This was sufficient to silence the interneuron for the duration of the current injection. In the first simulation, silencing was centered on the peak of the place field input. 1050 laps were simulated, and the data were then pooled and averaged to generate *Figure 5—figure supplement 2A,B* (see *Royer et al. (2012)* for details of data analysis). In the second simulation , silencing was centered on a random location within 20 cm of the peak of the place field input. We performed this additional simulation to account for the fact that the silencing was centered on a fixed portion of the track in the protocol of Royer and colleagues, so that the

place cells analyzed typically were only silenced over part of their place field. We again simulated 1050 laps, with silencing centered on a different location in each lap, and then pooled and averaged the resulting data (*Figure 5—figure supplement 2C,D*).

## Dorsoventral changes

To model changes in theta dynamics along the dorsoventral axis, we simultaneously varied the place field width $\sigma$, noise $\sigma_n^E$, excitatory synaptic weight $w^E$, depolarizing current to interneurons $I_0^I$ and the pacemaker drive $I_\theta$. For *Figure 4A,B*, only $I_0^I$ was varied and all other parameters were as above. For *Figure 4C,D*, we chose two parameter sets representing the dorsal and ventral poles. For the dorsal pole, the parameters were: $\sigma = 45$ cm; $\sigma_n^E = 0.7$ mV; $I_0^I = 80.455$ pA; $I_\theta = 1.95$ pA; $w^E = 0.53$ nS. For the ventral pole, the parameters were: $\sigma = 600$ cm; $\sigma_n^E = 3$ mV; $I_0^I = 79.525$ pA; $I_\theta = 0.12$ pA; $w^E = 0.081$ nS. In both cases, the running speed was set to $v = 30$ cm/s.

## Calculation of precession frequency

To estimate the theta frequency of the simulated neurons, the membrane potential was bandpass filtered at $6.25 - 10$ Hz and the instantaneous phase was calculated via a Hilbert transform. The phase was unwrapped and then smoothed using a moving average of width 250 ms. The gradient was calculated at each time point to obtain the instantaneous frequency.

To determine the precession frequency at different running speeds, we calculated the average membrane frequency within a radius of 15 cm around the place field center on each pass through the place field. To remove artefactual frequency estimates arising due to the bursting dynamics within theta cycles, we excluded individual runs based on the variability of the instantaneous place cell frequency within this 15 cm radius. Specifically, we excluded runs on which the standard deviation was greater than 1.75 times the mean standard deviation over all runs at that speed. This excludes cases in which the estimated frequency fluctuated rapidly on a short timescale.

## Analysis of phase precession statistics

To estimate the strength of phase precession in each pyramidal cell (the *single-cell phase precession metric*), we calculated the Pearson correlation between the vector of spike phases $\Phi$ and the vector of the animal's location $X$ at the time of each spike on a single lap. The phase offset was chosen in order to minimize this correlation, i.e. to obtain the most negative possible (strongest) correlation between spike phase and the animal's location (*Foster and Wilson, 2007*; *Feng et al., 2015*). Specifically, given the vectors $X$ and $\Phi$, we calculated the correlation $\rho(X, \Phi + \widetilde{\phi})$, where $\widetilde{\phi} = \mathrm{argmin}_\phi(\rho(X, \Phi + \phi))$. This metric was also used to measure single-cell phase precession pooled over multiple laps (*Figure 7—figure supplement 4*).

To obtain the measure of population phase precession (the *population phase precession metric*), we pooled the spikes of all pyramidal cells on a single lap. We again calculated the correlation between the vector of pooled spike phases $\Phi_{\mathrm{pop}}$ and the vector whose entries are given by the distance of the animal from the place field center of the corresponding cell in the pooled spike phase vector at the time of that spike $X_{\mathrm{pop}}$. As for the single cell case, the phase offset was chosen in order to minimize this correlation by calculating $\rho(X_{\mathrm{pop}}, \Phi_{\mathrm{pop}} + \widetilde{\phi}_{\mathrm{pop}})$, where $\widetilde{\phi}_{\mathrm{pop}} = \mathrm{argmin}_{\phi_{\mathrm{pop}}}(\rho(X_{\mathrm{pop}}, \Phi_{\mathrm{pop}} + \phi_{\mathrm{pop}}))$.

To measure the strength of sequential activity in the population (the single-cycle theta sequence metric), we analyzed the data on a cycle-by-cycle basis. For each cycle, we calculated the Pearson correlation between the vector of spike times in the population and the vector whose entries are given by the place field center corresponding to each spike in this first vector. Theta windows for this method had a temporal width equal to the period of the pacemaker input to the network. The offset of theta windows was given by the phase offset $\widetilde{\phi}_{\mathrm{pop}}$ which maximized the population phase precession measure for that lap. This allows for the possibility of an offset between the simulated CA1 network theta activity and the septal input oscillation.

## Place field mapping

For network simulations, the number of simulated pyramidal cells was held constant ($N_p = 1000$) and the number of interneurons was varied. This choice was made to avoid changes in correlation values

introduced by changes in sample size. The number of interneurons was always chosen to be a divisor of the number of pyramidal cells so that there was an equal number of place cells for each interneuron. Each simulated place cell was given exactly one place field. For random place field mapping, place field locations were generated by a uniform distribution over a linear track. For optimal place field mapping, place field locations were defined so that the place cells associated with a single interneuron were equally spaced along the track and so that the entire population of place cells uniformly covered the track.

## Reduced model of remapping with non-overlap constraint

Here we quantify the capacity of the network under the assumption that pyramidal cells which couple to the same interneuron cannot have overlapping place fields. We use three distinct measures of the network capacity: the number of spatial maps at a given spatial acuity; the number of cell assemblies; the number of phase sequences. These derivations were used to provide the capacity estimates stated in the main text.

### Number of distinct spatial maps

To determine the number of spatial maps available to the network, we considered a simplified model in which each place cell can map to a set of discrete locations on a linear track of length $L$. Specifically, the track is divided into equal bins of size $x_{\mathrm{res}} = L/N_{\mathrm{bins}}$, where $N_{\mathrm{bins}}$ is the number of bins and $x_{\mathrm{res}}$ determines the spatial resolution of the place map. To avoid finite size effects, we assume periodic boundary conditions (i.e., a circular track). The number of place fields to be mapped onto the track depends on both the number of place cells $N_p$ and the average number of place fields per place cell $F$ (which can be greater or less than one). Given a number of interneurons $N_I$, the population of $N_p$ place cells is divided into $N_I$ equal subsets, so that each interneuron is associated with the same number of place cells. We assume that there is an exclusion zone of size $D$ which sets the minimum distance for which place cells associated with the same interneuron can be mapped, so that $N_d = D/x_{\mathrm{res}}$ is the minimum separation in terms of the number of bins. In general, multiple cells may map to the same bin, or no cells may map onto a given bin, provided that the non-overlap constraint is obeyed.

We can then consider the number of ways in which $FN_p$ place fields can be mapped onto the track without violating this constraint. We can calculate this number by counting the number of possible choices for each for each place field $i$, where $1 \le i \le FN_p$. For the first place field $i = 1$, there are $N_p N_{\mathrm{bins}}$ possible choices, since we can choose from $N_p$ place cells and $N_{\mathrm{bins}}$ spatial locations. For the next choice $i = 2$, there are $N_p N_{\mathrm{bins}} - N_d N_p/N_I$ choices, due to the exclusion zone about the first cell, which excludes $N_p/N_I$ cells from being mapped onto $N_d$ of the possible bins. In general, there are $N_p N_{\mathrm{bins}} - (i-1)N_d N_p/N_I$ for the $i$th choice. Hence, the total number of combinations is:

$$N = \prod_{i=1}^{FN_p} \left( N_p N_{\mathrm{bins}} - (i-1)\frac{N_p}{N_I}N_d \right) \tag{24}$$

which can be simplified by noting that $N_d = N_{\mathrm{bins}}D/L$ so that:

$$N = \left( N_p N_{\mathrm{bins}} \right)^{FN_p} \prod_{i=1}^{FN_p} \left( 1 - (i-1)\frac{D}{LN_I} \right) \tag{25}$$

The above analysis gives the number of ordered choices of place cells and spatial bins, but overcounts the number of distinct maps by allowing the same map to be obtained through multiple choice sequences. This can be corrected by a factor of $(FN_p)!$ to obtain the number of distinct maps:

$$N_{\mathrm{maps}} = \frac{\left( N_p N_{\mathrm{bins}} \right)^{FN_p}}{(FN_p)!} \prod_{i=1}^{FN_p} \left( 1 - (i-1)\frac{D}{LN_I} \right) \tag{26}$$

Taking the logarithm and applying Stirling's approximation gives:

$$\log N_{\mathrm{maps}} \approx FN_p(1 + \log L - \log x_{\mathrm{res}} - \log F) + \sum_{i=1}^{FN_p} \log\left( 1 - (i-1)\frac{D}{LN_I} \right) \tag{27}$$

This equation was used to determine the number of spatial maps under which coherent theta sequences can be generated, as stated in the main text.

## Number of cell assemblies

In addition to analyzing the number of spatial maps available to the network, we also considered the number of distinct cell assemblies which can be generated by the network without causing disruption. We define a cell assembly to be a set of coactive cells. We now calculate the number of cell assemblies which contain $n$ place cells under the non-overlap constraint. To construct a cell assembly satisfying the constraint, it is sufficient to simply select $n$ distinct interneurons and then select a place cell associated with each interneuron. The number of possible cell assemblies $N_{\mathrm{CA}}$ is therefore:

$$N_{\mathrm{CA}} = \binom{N_I}{n}\left(\frac{N_P}{N_I}\right)^n; \qquad n \leq N_I \tag{28}$$

As before, we can simplify this using Stirling's approximation:

$$\log(N_{\mathrm{CA}}) \approx N_I \log N_I - (N_I - n)\log(N_I - n) + n\left(\log N_p - \log N_I - \log n\right) \tag{29}$$

We used this equation to estimate the number of cell assemblies that can be expressed in the network under the parameter assumptions stated in the main text.

## Number of phase sequences

Finally, we considered how many phase sequences the network can generate without introducing disruption. A phase sequence is defined as an ordered set of cell assemblies (*Hebb, 1949*). We assume that a phase sequence is a discrete sequence of $m$ cell assemblies, and that no two cells in a phase sequence can couple to the same interneuron. A phase sequence can then be constructed by repeatedly constructing cell assemblies as above, where the available interneurons for each subsequent cell assembly are given by those not already selected in previous assemblies within the sequence. The number of phase sequences $N_{\mathrm{PS}}$ is then:

$$N_{\mathrm{PS}} = \prod_{i=1}^{m}\binom{N_I - (i-1)n}{n}\left(\frac{N_P}{N_I}\right)^n; \qquad n \leq \frac{N_I}{m} \tag{30}$$

which can be approximated as:

$$\log N_{\mathrm{PS}} \approx \sum_{i=1}^{m}\left[(N_I - (i-1)n)\log(N_I - (i-1)n) - (N_I - in)\log(N_I - in) + n\left(\log N_p - \log N_I - \log n\right)\right] \tag{31}$$

Again, this equation was used to estimate the capacity of the network to generate distinct sequences as stated in the main text.

## Acknowledgements

We would like to thank Mark Brandon for helpful comments on the manuscript. This work was supported by EPSRC, BBSCR and MRC.

## Additional information

**Competing interests**

MCWvR: Reviewing editor, *eLife*. The other authors declare that no competing interests exist.

**Funding**

| Funder | Grant reference number | Author |
| --- | --- | --- |
| Engineering and Physical Sciences Research Council | EP/F500385/1 | Angus Chadwick Mark CW van Rossum |
| Biotechnology and Biological Sciences Research Council | BB/L010496/1 | Matthew F Nolan |

The funders had no role in study design, data collection and interpretation, or the decision to submit the work for publication.

## Author contributions

AC, Conception and design, Acquisition of data, Analysis and interpretation of data, Drafting or revising the article; MCWvR, MFN, Conception and design, Analysis and interpretation of data, Drafting or revising the article

## Author ORCIDs

Angus Chadwick, http://orcid.org/0000-0003-2664-0746
Mark CW van Rossum, http://orcid.org/0000-0001-6525-6814
Matthew F Nolan, http://orcid.org/0000-0003-1062-6501

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
