## [Decision Letter]

Thank you for submitting your article "Flexible theta sequence compression mediated via phase precessing interneurons" for consideration by *eLife*. Your article has been favorably evaluated by Timothy Behrens (Senior Editor) and three reviewers, one of whom is a member of our Board of Reviewing Editors.

The reviewers have discussed the reviews with one another and the Reviewing Editor has drafted this decision to help you prepare a revised submission.

In this thorough and clearly written paper, Chadwick et al. present a novel mechanistic model framework for theta sequences. They advance a model for phase precession in which interneurons receive theta-frequency inputs and pyramidal cells receive a driving Gaussian envelope. Their model distinguishes itself from others by having interneurons (PV) play a critical role. Specifically, the theta rhythm is input via medial septal GABAergic input to interneurons that are coupled with pyramidal (place) cells. This is different from any existing models (as nicely summarized in a table overview (Figure 1—figure supplement 1). The authors describe experimental phenomenon that have been traditionally difficult for other models to capture.

The authors use a reduced model to motivate, constrain and understand their model framework and so find and explain how it can capture the range of results from experimental work regarding theta sequences (e.g., theta waves propagating along dorsoventral axis, speed dependence etc.).

In the end, they show (via Figure 8) and describe their model mechanism in operation with STDP aspects, and have set down core (ancillary and network topographic) predictions.

The model has the potential to be quite interesting to researchers interested in temporal coding in the hippocampus. Currently there is no universally accepted model of phase precession that seems to explain all the data out there, which suggests either the right one has not yet been proposed or multiple mechanisms are at play. Either way, this model has the potential to be thought-provoking and to drive additional research that will be required to resolve the mechanisms underlying phase precession in hippocampus and associated structures.

Overall, they seem to have captured an "essential skeleton" for phase precession based on dynamically integrating pacemaker and spatial signals via inhibitory interneurons. It is especially interesting that this model mechanism could potentially be used as a base on which to build other existing mechanisms that include more details and biophysics (e.g., dendritic and somatic interference in pyramidal cells).

The reviewers would appreciate consideration of the following points in a revised submission.

Major points:

1) I would have guessed that this model relies specifically on the tuning of the strengths of the interaction between the pyramidal cell and the interneurons and also (related) the specific timing of the interactions between the two cells. I would guess that the model would be rather sensitive to any noise in the system. Perhaps the authors could demonstrate how phase precession is impacted by extraneous synaptic input noise?

2) Looking at Royer et al. (2012), the PV-interneurons seem to fire strongly over the entire maze (as do the non-PV interneurons). My understanding is that this is consistent with experimental findings reported from other laboratories. The model presented here requires that interneurons be coupled to pyramidal cells with mostly non-overlapping place fields. While I am satisfied that it is possible for the circuit to be configured in this way (and the authors spend a lot to space going through the combinatorics), I think it is unlikely that every environment would result in the pyramidal and interneuron pairs being organized in this way. Still, not all putative pyramidal cells show strong phase precession. I was curious if the authors might be able to make some prediction about the distribution of phase precession "strengths" that would result from their model with randomly assigned place fields and/or pyramidal cell-interneuron couplings.

3) In Figure 5—figure supplement 2, the authors claim to have replicated the results of Royer et al. 2012. To me it seems that the PV-inactivation accelerates the phase precession, whereas Royer reported a delay in precession. Put another way, the shifts reported by both groups seem to go in opposite directions from each other.

---

## [Author Response]

*Major points:*

*1) I would have guessed that this model relies specifically on the tuning of the strengths of the interaction between the pyramidal cell and the interneurons and also (related) the specific timing of the interactions between the two cells. I would guess that the model would be rather sensitive to any noise in the system. Perhaps the authors could demonstrate how phase precession is impacted by extraneous synaptic input noise?*

To address this question, we have performed additional simulations in which we inject varying levels of extraneous noise into the interneurons or pyramidal cells in the model (Figure 7—figure supplement 2 in the revised manuscript). We find that the phase precession and sequences continue to be robustly generated when several millivolts of noise is injected into pyramidal cells. Interneuron noise has a similar effect to the disruptive place field overlap we considered previously, causing a disruption of sequences but not single-trial pyramidal cell phase precession. In contrast, the disruption observed when large amplitude noise is injected into pyramidal cells involves a decrease in the strength of single-trial pyramidal cell phase precession as well as population sequences. These findings are now described in the second paragraph of the subsection “Network reconfiguration can dissociate single-trial phase precession and theta sequences”.

*2) Looking at Royer et al. (2012), the PV-interneurons seem to fire strongly over the entire maze (as do the non-PV interneurons). My understanding is that this is consistent with experimental findings reported from other laboratories. The model presented here requires that interneurons be coupled to pyramidal cells with mostly non-overlapping place fields. While I am satisfied that it is possible for the circuit to be configured in this way (and the authors spend a lot to space going through the combinatorics), I think it is unlikely that every environment would result in the pyramidal and interneuron pairs being organized in this way. Still, not all putative pyramidal cells show strong phase precession. I was curious if the authors might be able to make some prediction about the distribution of phase precession "strengths" that would result from their model with randomly assigned place fields and/or pyramidal cell-interneuron couplings.*

We have now analysed the distribution of phase precession strengths in a population of pyramidal cells with random maps of various densities (Figure 7—figure supplement 4 in the revised manuscript). As suggested by the reviewer, we find that a substantial fraction of the pyramidal cell population continues to exhibit robust phase precession despite disruption in the network.

With just one pyramidal cell per interneuron, all pyramidal cells exhibit robust phase precession, and cell to cell variability in phase precession strength reflects only sample size effects. With increasing numbers of pyramidal cells per interneuron, the mean phase precession strength in the population steadily decreases, but a substantial proportion nevertheless continue to exhibit strong phase precession even with 10 active pyramidal cells per interneuron on a 5 meter track. These findings are now described in the third paragraph of the subsection “Theta sequences can be generated in a large number of spatial maps”.

*3) In Figure 5—figure supplement 2, the authors claim to have replicated the results of Royer et al. 2012. To me it seems that the PV-inactivation accelerates the phase precession, whereas Royer reported a delay in precession. Put another way, the shifts reported by both groups seem to go in opposite directions from each other.*

For the simplest stimulation protocol we implemented (Figure 5—figure supplement 2 and B in the original manuscript, Figure 5—figure supplement 3A and B in the revised manuscript), phase precession does shift in the same direction reported by Royer et al. (compare this figure to their Figure 7). However, the reviewer is correct that when we tested a second, more detailed stimulation protocol, the shift appears to go in the opposite direction, as well as decreasing in magnitude. As we noted in the original manuscript, theta coordination of pyramidal cells is entirely absent during periods of interneuron inactivation in the model. Thus, while we do observe a phase shift in the averaged data, this is actually caused by a shift in the firing frequency of the pyramidal cell during interneuron silencing. The details of how this frequency shift translates into a phase shift in the averaged data are sensitive to specific details of the stimulation protocol, such as the timing and duration of inactivation within the place field. Thus, while our model can reproduce both the magnitude and the direction of the phase shifts reported by Royer et al., our results suggest that the phase shift they reported does not reflect the fundamental role of PV neurons in generating pyramidal cell phase precession. We have now clarified this point in the revised manuscript (subsection “Robust phase precession is generated along two-dimensional trajectories”, third paragraph).